# SMuCo: Reinforcement Learning for Visual Control via Sequential Multi-view Total Correlation

**Tong Cheng**[*1]  **Hang Dong**[2]  **Lu Wang**[2]  **Bo Qiao**[2]  **Qingwei Lin**[2]  **Saravan Rajmohan**[3]  **Thomas Moscibroda**[4]

[1]College of Computing and Data Science , Nanyang Technological University , Singapore, 639798 ,
[2]Microsoft AI , Beijing, China, 100000
[3]Microsoft 365 , Beijing, China, 100000
[4]Microsoft Azure , Beijing, China, 100000

## Abstract

The advent of abundant image data has catalyzed the advancement of visual control in reinforcement learning (RL) systems, leveraging multiple viewpoints to capture the same physical states, which could enhance control performance theoretically. However, integrating multi-view data into representation learning remains challenging. In this paper, we introduce SMuCo, an innovative multi-view reinforcement learning algorithm that constructs robust latent representations by optimizing multi-view sequential total correlation. This technique effectively captures task-relevant information and temporal dynamics while filtering out irrelevant data. Our method supports an unlimited number of views and demonstrates superior performance over leading model-free and model-based RL algorithms. Empirical results from the DeepMind Control Suite and the Sapien Basic Manipulation Task confirm SMuCo's enhanced efficacy, significantly improving task performance across diverse scenarios and views.

## 1 INTRODUCTION

The challenge of visual control or learning from pixels entails addressing a reinforcement learning (RL) problem where states are represented in the form of images. Extensive investigations into this problem have been conducted in prior studies, as noted in works such as [Kirk et al., 2021], showcasing commendable performance on continuous control tasks by directly utilizing images as input. Despite these achievements, the performance in visual control problems lags behind that of works employing physical states as input for direct control, as demonstrated in [Yarats et al., 2019]. This disparity primarily arises from the challenge of ef-

fectively extracting all *task-relevant* information while filtering out *task-irrelevant* details during the representation learning process, as highlighted in [Zhang et al., 2021, Fan and Li, 2022]. Notably, with the increasing availability of multi-view data in various application scenarios, additional perspectives now contribute to distinguishing task-relevant information from task-irrelevant information, independent of specific actions. In the context of the robot arm catching problem, obtaining images from both upper and horizontal viewpoints offers valuable insights. By conducting a comparative analysis of images from these two perspectives, relevant information about the running robot arm can be extracted, effectively isolating it from background elements that are irrelevant to the control task, as emphasized in [Xiang et al., 2020]. This highlights the pressing need to develop a mechanism that enhances visual control performance through skilled representation learning from multi-view data.

An additional coveted attribute for the learned representation in the visual control task of reinforcement learning is its ability to encompass the predictability of future states based on potential actions. At the same time, it should discard task-irrelevant visual details, thereby capturing the temporal structure of task-relevant dynamics, as discussed in [Fan and Li, 2022]. This learned representation not only enhances the robustness of the acquired policy in unfamiliar environments but also addresses challenges stemming from the complexity of high dimensionality and the causal confusion effect, as outlined in [de Haan et al., 2019], which arises from task-irrelevant information.

This study focuses on representation learning for visual control tasks utilizing input images from multiple views, introducing a novel reinforcement learning algorithm named SMuCo. Under the multi-view setting [Federici et al., 2020, Li et al., 2016, Fischer, 2020], where shared information among multi-view observations is considered task-relevant and unshared information is considered task-irrelevant, our proposed method adeptly learns task-relevant temporal dynamics while discarding extraneous information for visual

---

[*]This work is done during internship in Microsoft Research.

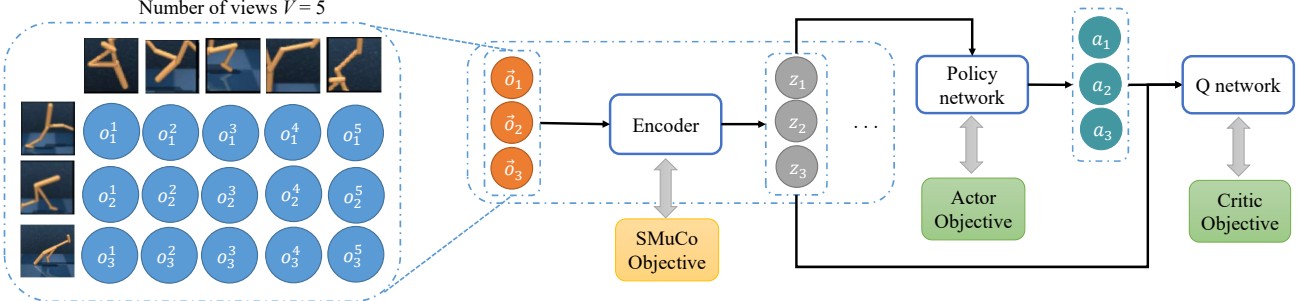

Figure 1: RL with **S**equential **Mu**lti-view Total **Co**rrelation (SMuCo) framework takes multi-view observations over $T$ time steps as input to learn complete representations for downstream RL tasks. The dimension of each observation $\vec{o}_i$ ($i = 1, 2, 3$) equals to the number of views $V$. The SMuCo objective is derived as a lower bound of sequential total correlation between multi-view observation sequences and representations sequences.

control tasks. In our framework, depicted in Figure 1, diverse observation viewpoints are encoded into a unified representation through deep neural networks, with this learned representation serving as the state for training the policy through reinforcement learning. To train the encoder, we formulate the SMuCo objective, akin to sequential total correlation for sequences of multi-view observations. This objective guides the learning process, emphasizing the preservation of task-relevant temporal dynamics and the elimination of task-irrelevant information.

Our contributions of this work are summarized as follows:

- We propose SMuCo, a novel reinforcement learning framework for representation learning from multiple views in visual control problems based on multi-view total correlation.

- We derive the SMuCo objective that represents the multi-view total correlation between sequential observations and representations in SMuCo to learn representations that can well capture task-relevant temporal dynamics while discarding task-irrelevant information.

- We empirically validate that SMuCo can learn a sufficient and concise representation from multiple views of images by demonstrating that SMuCo achieves higher scores than both model-free and model-based state-of-the-art (SOTA) RL algorithms on a number of multi-view image-based control tasks.

## 2 RELATED WORK

**Visual Control in Reinforcement Learning**. Various efforts have been undertaken to develop robust representations for visual control tasks in reinforcement learning. Some approaches address the visual control problem through contrastive viewpoints, as seen in CURL, which utilizes contrastive learning to enhance agent robustness and generalization [Laskin et al., 2020b]. Contrastive Predictive Coding

(CPC) proposes learning representations by predicting the future latent space [van den Oord et al., 2018]. Additionally, there are works grounded in the theory of bisimulation, such as DBC [Zhang et al., 2021], a bisimulation-based reinforcement learning algorithm aiming to extract state information to eliminate redundancy in natural video input. PSE (Policy Similarity Embedding) [Agarwal et al., 2021] and DBC-IR-ID [Kemertas and Aumentado-Armstrong, 2021] represent improved versions of DBC, with DBC-IR-ID incorporating constraints in the representation space, intrinsic rewards, and inverse dynamics. Furthermore, some works design reinforcement learning algorithms using information-theoretic auxiliary tasks. Among these, DRIBO [Fan and Li, 2022] establishes an RL framework akin to CURL, leveraging the multi-view information bottleneck method [Federici et al., 2020]. PI-SAC [Lee et al., 2020b] utilizes a conditional entropy bottleneck (CEB) to predict future observations and rewards. However, most of these existing methods are not applicable to the multi-view setting. Notably, DRIBO can only handle situations with two views due to the pairwise formulation of the objective. Random PadResize and CycAug [Ma et al., 2023] are recently proposed data augmentation techniques to enhance the sample efficiency of visual reinforcement learning algorithms. In order to mitigate visual deadly triad, A-LIX [Cetin et al., 2022] provides adaptive regularization to the encoder's gradients to avoid self-overfitting. In environments characterized by partial observability, it is rational to employ multiple policies to handle diverse visual observations, as explored in [Shang and Ryoo, 2023]. However, in multi-view settings, increasing the number of policies with the growing number of views becomes impractical. Therefore, in the context of multi-view scenarios, the potential lies in mapping these observations into a comprehensive and condensed representation, initiating the learning process from the induced latent space. The challenges associated with learning from visual observations are further compounded by the instability of the Q function in off-policy RL algorithms, as discussed in

[Hansen et al., 2021]. This instability is primarily attributed to redundant information present in raw sensor observations. Nevertheless, the transformation of these raw observations into sufficient and compact representations through multi-view total correlation, as introduced in [Federici et al., 2020], can mitigate this issue. The significance of diverse visual perspectives on the learning and generalization performance in the context of visual control is underscored in [Hsu et al., 2022], emphasizing the non-negligible impact that different viewpoints can have on the overall effectiveness of the learning process.

**Multi-view Representation Learning**. Research in multi-view representation learning has delved into extracting robust and concise representations from multi-view data, offering various network architectures to transform such data into representations with desirable properties. The Memory Fusion Network (MFN) [Zadeh et al., 2018] constructs sequential multi-view representations, incorporating accountability for interactions within a neural architecture. The Multi-view Laplacian Network [Huang et al., 2021] is designed to learn spectral representations with consensus from multi-view data. CPM-Nets [Zhang et al., 2019] aim to learn a comprehensive representation of multi-view data, accommodating potential partialities. Furthermore, appropriate loss functions are crucial for training deep networks to instill ideal properties like sample efficiency and robustness into representations. S2R [Yang et al., 2022] is a multi-view reinforcement learning algorithm that extends the two-view conditional entropy bottleneck method to a multi-view setting, facilitating the learning of sample-efficient representations. DRIBO [Fan and Li, 2022] utilizes the mutual predictability of multi-view (pairwise) observations to acquire a robust representation devoid of task-irrelevant information. However, these methods cannot be directly applied to visual control problems due to the absence of considerations for temporal predictability. Fuse2Control (F2C) [Hwang et al., 2023] is an information-theoretic Multi-View Reinforcement Learning framework that learns a latent state space model. It is good at handling missing view problem. However, the temporal length F2C considers is limited, which could hinder the learning process of sequential decision making task.

**Total Correlation**. Total correlation, a fundamental method derived from information theory, plays a pivotal role beyond representation learning within the realm of AI, which has been applied across a diverse spectrum of tasks ([Chen et al., 2018, Locatello et al., 2019, Kim and Mnih, 2018]). The Total Correlation method is designed to capture shared information among data with minimal sufficiency, focusing on the independence among random variables [Watanabe, 1960]. In the context of independent component analysis (ICA), [Cardoso, 2003] introduces a comprehensive framework that establishes connections between mutual information, entropy, and non-Gaussianity, all without relying on decorrelation constraints. This framework contributes significantly to understanding the underlying structures within complex datasets by leveraging the inherent dependencies among variables. For the domain of structure discovery, [Ver Steeg and Galstyan, 2014] proposes a novel methodology centered around learning a hierarchical structure of progressively abstract representations of intricate data sets. This approach is underpinned by optimizing an information-theoretic objective, ensuring that the learned representations capture meaningful and salient features of the data while facilitating interpretability and scalability. The Total Correlation Explanation (CorEx) principle has been leveraged in unsupervised learning to enhance interpretability. Total correlation, as discussed in [Gao et al., 2019b], plays a role in characterizing disentanglement and dependence within representations. MVTC [Hwang et al., 2021] introduces an information-theoretic approach to transform multi-view data into complete and minimally sufficient representations. These works collectively highlight the versatility and power of total correlation as a foundational concept in information theory, showcasing its applicability across a wide range of AI applications.

# 3 SMUCO

In this section, we give the definitions and approximate formulation of sequential total correlation in SMuCo that can capture the task-relevant information and temporal dynamics while discarding task-irrelevant information in the learned representations for visual control tasks, as well as the visual control RL algorithm based on SMuCo.

## 3.1 MULTI-VIEW TOTAL CORRELATION

Total correlation, also referred to as multivariate mutual information, has been shown to be able to characterize informativeness and disentanglement from observations [Gao et al., 2019a]. Optimizing total correlation can guide the stochastic search process for a set of latent factors that explain best the correlations in the original data [Steeg and Galstyan, 2014]. Under the multi-view setting, the total correlation between multi-view observations and representation is defined as:

$$TC(\vec{O}; Z) = TC(\vec{O}) - TC(\vec{O} \mid Z) \qquad (1)$$

which can be rewritten into:

$$TC(\vec{O}; Z) = \sum_{v=1}^{V} I(O^v; Z) - I(\vec{O}; Z) \qquad (2)$$

where $I$ denotes mutual information, and $V$ is the number of viewpoints for observations. Maximizing the expected total correlation between multi-view observation and representation can not only enforce informativeness but also guarantee sufficiency of the representation [Hwang et al., 2021].

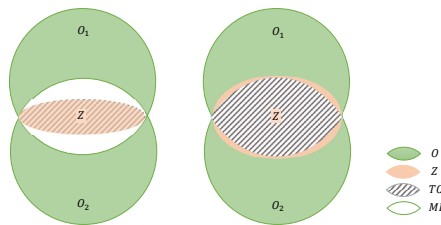

Figure 2: Illustration of total correlation on two views.

In unsupervised representation learning, total correlation has been used to obtain complete and minimal sufficient representations from multiple views [Hwang et al., 2021]. Figure 2 provides an illustration of the intuition behind such a mechanism with a simple example of two views. In Figure 2, the green circles denote the entropy of the observations $O$ and the red ellipses denote the entropy of the representation $Z$. According to the assumptions of the multi-view setting, these two views have overlapping information, whose entropy is denoted as the white area. According to Equation 2, the value of TC is equal to the shaded area in the figure, which is equal to the entropy of $Z$ in the left part. From left to right, the total correlation has increased, and the representation is encouraged to incorporate more shared information between the two views thus can extract more task-relevant information under the multi-view setting.

### 3.2 SEQUENTIAL MULTI-VIEW TOTAL CORRELATION

Temporal structure is important in sequential decision making problems and representations incorporating temporal dynamics have better predictability of future states. To accurately identify temporal dynamics and remove task-irrelevant information from learned representations for visual control tasks, the encoder should be able to correlate sequential observations and representations in the temporal structure [Fan and Li, 2022].

Empirically, the success of related works such as DRIBO [Fan and Li, 2022] and PI-SAC [Lee et al., 2020b] have demonstrated the advantage of considering this temporal structure. For visual control problems with multi-view data, we extend the formulation of total correlation to sequences of multi-view observations conditioned on the action sequences of the MDPs, motivated by the success of PI-SAC [Lee et al., 2020b]. PI-SAC is a model-free reinforcement learning algorithm that learns compressive representations of predictive information to improve sample efficiency. It can capture the temporal dynamics of the environment into the learned representation by substituting random variables in CEB [Fischer, 2020]

with a combination of sequences of previous and future observations, actions, and rewards. Specifically, CEB aims to optimize the following objective:

$$CEB \equiv \min_Z \beta I(X; Z \mid Y) - I(Y; Z) \quad (3)$$

According to [Lee et al., 2020b], it follows that

$$CEB \leq \mathbb{E}_{x,y,z \sim p(x,y)e(z|x)} \beta \log \frac{e(z \mid x)}{b(z \mid y)} - I(Y; Z) \quad (4)$$

where $e(z \mid x)$ is the true encoder distribution representation $z$ comes from and $b(z|y)$ is the variational backwards encoder distribution that approximates the unknown true distribution $p(z \mid y)$. The minimization of CEB can be approximated by minimization of this upper bound. In PI-SAC, the loss function after substitution of CEB has the following form:

$$
\begin{aligned}
\mathcal{L} = \mathbb{E} \log \frac{e\left(z_0 \mid o_{-T+1:0}, a_{0:T-1}\right)}{b\left(z_0 \mid s_{1:T}, r_{1:T}\right)} \\
+ \log \frac{b\left(z_0 \mid o_{1:T}, r_{1:T}\right)}{\frac{1}{K} \sum_{k=1}^{K} b\left(z_0 \mid o_{1:T}^k, r_{1:T}^k\right)} \quad (5)
\end{aligned}
$$

where expectation is taken over $(o_{-T+1:T}, a_{0:T-1}, r_{1:T} \sim \mathcal{D}, z_0 \sim e(z_0 \mid \cdot))$.

Adopting this idea of PI-SAC, we show our extension of multi-view total correlation to sequential multi-view total correlation (SMTC) as follows. The SMTC of a sequence of observations and representations is defined as:

$$
\begin{aligned}
SMTC(\vec{O}_{1:T}; Z_{1:T} \mid A_{1:T}) = \\
\sum_{v=1}^{V} I(O_{1:T}^v; Z_{1:T} \mid A_{1:T}) - I(\vec{O}_{1:T}; Z_{1:T} \mid A_{1:T}) \quad (6)
\end{aligned}
$$

where $\vec{O}_{1:T}$ denotes the sequence of the observation view vectors, each with $V$ views, $A$ denotes actions, $Z$ denotes the representation, and $T$ denotes the sequence length. According to [Lee et al., 2020b, Mazoure et al., 2020], the encoder predicts future states more accurately under the condition of multiple future actions. Maximizing the above SMTC is equivalent to maximizing $\sum_{v=1}^{V} I(O_{1:T}^v; Z_{1:T} \mid A_{1:T})$ and minimizing $I(\vec{O}_{1:T}; Z_{1:T} \mid A_{1:T})$. The former term makes the obtained representation complete as it encourages $Z$ to be informative, while the latter term guarantees the conciseness of the resulted representation. Therefore, the maximization of SMTC enforces the representation to capture minimally sufficient correlations among different views over the sequences. Unfortunately, the calculation of both of these terms requires the calculation of mutual information among random vectors, which is notoriously difficult to compute [Belghazi et al., 2018, Fan and Li, 2022]. Therefore, we instead try to find an appropriate surrogate for this SMTC objective.

Let $O_{1:T}$ be observation sequence and $Z_{1:T}$ be representation sequence, whose joint distribution is $p(O_{1:T}, Z_{1:T}) = \prod_{t=1}^{T} p(O_t, Z_t \mid O_{t-1}, Z_{t-1}, A_{t-1})$ where $A_{1:T}$ is action sequence and $p(O_1, Z_1 \mid O_0, Z_0, A_0) = p(O_1, Z_1)$. Let

$\vec{O}_{1:T}$ be multi-view observation sequence with dim $\vec{O} = V$ and temporal length $T$. We derive a tractable lower bound of sequential multi-view total correlation as follows:

**Theorem 3.1.** *The sequential total correlation between sequences of multi-view observation and representation on condition of action sequence has the following lower bound:*

$$SMTC(\vec{O}_{1:T}; Z_{1:T} \mid A_{1:T}) \geq$$

$$\sum_{v=1}^{V} \sum_{t=1}^{T} [H(O_t^v \mid Z_{t-1}, A_{t-1})$$

$$+ \mathbb{E}_{p(z_t, o_t^v \mid z_{t-1}, a_{t-1})} \ln q_\psi^v(o_t^v \mid z_t, z_{t-1}, a_{t-1})]$$

$$- \sum_{t=1}^{T} \sum_{s=1}^{T} \mathbb{E}_{p(\vec{o}_s)} [D_{KL}(p(z_t \mid o_s, \iota) \parallel r_\phi(z_t \mid \iota))] \quad (7)$$

*where $H$ is the entropy function, $\iota = (\vec{o}_{1:s-1}, z_{1:t-1}, a_{1:T})$, prior distribution $r_\phi(z_t) \approx p(z_t)$ is an approximate distribution for $\phi$, and posterior distribution $q_\psi(o_t^v \mid z_t, z_{t-1}, a_{t-1}) \approx p(o_t^v \mid z_t, z_{t-1}, a_{t-1})$ is an approximate distribution for $\psi$.*

Using this result, we construct the loss function for representation learning based on Equation 7 in SMuCo, which is detailed in Section 3.3. Proof of Theorem 3.1 is elaborated in the appendix.

### 3.3 VISUAL CONTROL WITH SMUCO

In the following, we show how the visual control task is resolved with our proposed SMuCo objective for representation learning. As shown in Figure 1, we use the SMuCo objective derived based on SMTC to learn the encoder, and the observations are encoded as states for the reinforcement learning part to learn the control policy. The details of the encoder are explained in the following part.

**Encoder**. According to Equation 7, terms on the right-hand side can be treated as three parts of the loss function of the encoder as follows:

$$\mathcal{L} = \mathcal{L}_{\text{REC}} + \mathcal{L}_{\text{LL}} + \mathcal{L}_{\text{TC}} \quad (8)$$

where the reconstruction entropy term $\mathcal{L}_{\text{REC}}$, the expected logarithmic likelihood term $\mathcal{L}_{\text{LL}}$ and the temporal contrastive term $\mathcal{L}_{\text{TC}}$ are defined as follows:

$$\mathcal{L}_{\text{REC}} = -\sum_{v=1}^{V} \sum_{t=1}^{T} H(O_t^v \mid Z_{t-1}, A_{t-1}), \quad (9)$$

$$\mathcal{L}_{\text{LL}} = -\sum_{v=1}^{V} \sum_{t=1}^{T} \mathbb{E}_{p_1} \ln q_\psi^v(o_t^v \mid z_t, z_{t-1}, a_{t-1}), \quad (10)$$

$$\mathcal{L}_{\text{TC}} = \sum_{t=1}^{T} \sum_{s=1}^{T} \mathbb{E}_{p_2} [D_{KL}(p(z_t \mid o_s, \iota) \parallel r_\phi(z_t \mid \iota))], \quad (11)$$

where $p_1 := p(z_t, o_t^v \mid z_{t-1}, a_{t-1})$ and $p_2 := p(\vec{o}_s)$.

For the benefits of the multi-view correlation, the completeness of multi-view representation is defined as the reconstruction ability of representation into each individual view [Hwang et al., 2021]. By minimizing the reconstruction entropy term $\mathcal{L}_{\text{REC}}$, we try to obtain the representation $Z$ which is a maximal compression of observation $O$, thus trying to eliminate irrelevant information from visual control tasks in the learned representation $Z$. Minimizing the expected log likelihood term $\mathcal{L}_{\text{LL}}$ conforms to the principle of maximizing log likelihood in statistical inference methods, trying to preserve the temporal dynamics of the sequence. $\mathcal{L}_{\text{TC}}$ is a regularization term for this surrogate loss function, preventing approximate prior distribution $r$ from divergence with true posterior distribution $p$.

**Joint Modeling**. We use Product of Expert (PoE) [Hinton, 2002] and Inverse Variance Weighted (IVW) [Cochran, 1954] for the joint modeling of multiple views. For each view, we assign a separate encoder and decoder network. After feeding multi-view observations into the encoder, a joint representation is obtained by aggregating $V$ separate representations into a single one, as illustrated in Figure 3. Reparamterization method [Kingma and Welling, 2014] is utilized to guarantee the feasibility of backpropagation over parameters of latent distributions.

Summing up, the training procedure of the encoder as well as the reinforcement learning policy is elaborated in Algorithm 1. We design our algorithm using a co-training paradigm, as updates of each component among encoder, actor, and critic require values passed through other components.

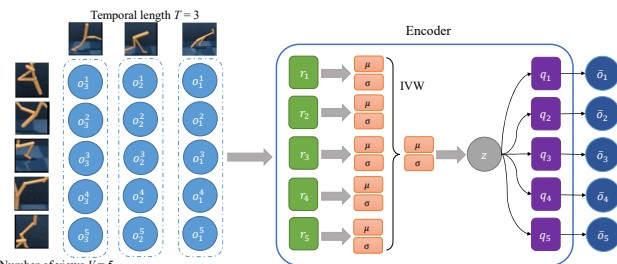

Figure 3: SMuCo encoder architecture. At each time step $i = 1, 2, 3$, the encoder takes one column of multi-view observations, i.e. $\vec{o}_i$, as input to generate a joint representation $z$.

## 4 EXPERIMENTS

### 4.1 EXPERIMENTAL SETUP

To assess the efficacy of our proposed method, we integrate SMuCo with SAC to tackle visual control tasks sourced from the DeepMind Control (DMC) Suite [Tassa et al., 2018] and Sapien [Xiang et al., 2020]. The experiments in

---

**Algorithm 1** SMuCo Training Procedure

---

**Input**: environment $E$, encoder $p$ parameters $(\phi, \psi)$, policy $\pi$ parameters $\theta$, Q function parameters $\eta_1, \eta_2$, replay buffer $\mathcal{D}$

1: Reset environment $E$ with multi-view observation $\vec{o}_0$.
2: Initialize representation $z_0 \sim p_\phi(\cdot \mid \vec{o}_0)$.
3: Initialize replay buffer $\mathcal{D}$.
4: Initialize target parameters: $\eta_{trgt,i} \leftarrow \eta_i$ where $i = 1, 2$.

5: **while** not convergence **do**
6:     Get action $a_t \sim \pi_\theta(\cdot \mid z_t)$.
7:     Get reward $r_t = R(\vec{o}_t, a_t)$.
8:     Get next observation $\vec{o}_{t+1} \sim \mathcal{P}(\cdot \mid \vec{o}_t, a_t)$.
9:     Get representation $z_{t+1} \sim p_\phi(\cdot \mid \vec{o}_{t+1})$.
10:    Push tuple $(\vec{o}_t, z_t, a_t, r_t, \vec{o}_{t+1}, z_{t+1})$ into replay buffer $\mathcal{D}$.
11:    **for** update steps **do**
12:       Update encoder $p$ by gradient descent using $\nabla_{\phi,\psi}\left(\mathcal{L}_{\text{REC}} + \mathcal{L}_{\text{LL}} + \mathcal{L}_{\text{TC}}\right)$ by sampling replay buffer $\mathcal{D}$.
13:       Update Q network by gradient descent using

$$\mathcal{L}_{Q_{\eta_i}} = \mathbb{E}_{B \subset \mathcal{D}}[Q_{\eta_i}(z_t, a_t) - y(r_t, z_{t+1})]^2$$

        where $(z_t, a_t, r_t, z_{t+1}) \in B$ and $i = 1, 2$.
14:       Update policy network $\pi$ by gradient descent using

$$\mathcal{L}_{\pi_\theta} = \mathbb{E}_{B \subset \mathcal{D}} \log \pi_\theta(a'_t \mid z_t) - \min_{i=1,2} Q_{\eta_i}(z_t, a'_t)$$

        where $a'_t \sim \pi_\theta(\cdot \mid z_t), z_t \in B$.
15:       Update target network by polyak averaging: $\phi_{\text{targ},i} \leftarrow \rho\phi_{\text{targ},i} + (1 - \rho)\phi_i$.
16:    **end for**
17: **end while**

---

this section are conducted on three servers, each equipped with 24 CPU cores, 110GB of memory, and NVIDIA Tesla A100 GPUs. The Sapien task involves multiple views obtained from cameras at various angles. In DMC tasks, where multi-view observations are not inherently available, we adopt random crop as the view generation method based on its reported advantages over other data augmentation methods, as highlighted in RAD [Laskin et al., 2020a]. Results are averaged across 5 random seeds, with each agent undergoing training for up to 500,000 steps. The network architectures and other hyperparameters are provided in detail in the appendix.

### 4.2 BASELINES

The following SOTA model-based and model-free methods are compared with our proposed methods: **Dreamerv2** [Hafner et al., 2021], **RAD** [Laskin et al., 2020a], **PI-SAC** [Lee et al., 2020b], **DrQ** [Yarats et al., 2021], **SLAC** [Lee et al., 2020a], and **DRIBO** [Fan and Li, 2022].

DreamerV2 sets itself apart by integrating a world model to understand agent behaviors, explicitly preserving latent dynamics. In contrast, other techniques like RAD and DrQ do not explicitly model dynamics. Both RAD and DrQ input transformed observations, raw pixels from interactions with the environment, into downstream reinforcement learning (RL) tasks. On the contrary, SMuCo considers the joint representation from encoding multi-view observations as the state, emphasizing task-relevant information for downstream RL tasks. In the case of RAD and DrQ, a notable distinction lies in observational transformation. RAD employs data augmentation techniques such as color jittering and random cropping, while DrQ samples transformation operators from an invariant state transformation set, referring to this approach as a data regularized method. Furthermore, PI-SAC achieves representation learning by maximizing a Conditional Entropy Bottleneck (CEB)-related surrogate as an auxiliary task for training the encoder. In contrast, DRIBO aims to maximize the mutual information between two marginal representations and the divergence of likelihood probability. These differences underscore the varied approaches and methodologies each method employs in the field of representation learning for RL.

While the training objective (Equation 8) of the encoder in SMuCo necessitates $V$ views over $T$ time steps, unlike other baselines with no such requirement, it doesn't introduce unfairness in performance evaluation. It's important to note that during the evaluation stage, the encoder parameters are frozen, and episodes are generated step by step in both SMuCo and other baselines. Although it may appear that SMuCo leverages information over a broader time window, the decision to utilize historical information is inherent to the design of the training objective. The availability of historical information is equal for both SMuCo and other baselines. However, SMuCo gives it more thoughtful consideration, leading to superior performance. In conclusion, as long as episodes are unrolled one step at a time during the evaluation stage, the comparison remains unbiased.

### 4.3 EVALUATION ON CONTROL TASKS

We evaluate SMuCo on tasks from DMC Suite [Tassa et al., 2018] and Sapien environment [Xiang et al., 2020] with other baselines mentioned above.

The experimental results of our proposed method compared with the baseline methods are shown in Figure 4, from which we can see that SMuCo achieves better performance than previous works including DrQ, RAD, Dreamerv2 and DRIBO, and comparable performance with PI-SAC and SLAC. In the cheetah run task, SMuCo converges faster than other baselines and achieves better performance than baselines except DreamerV2. Similarly, in the walker walk task, SMuCo converges faster than other baselines and achieves better performance than baselines except DRIBO. However,

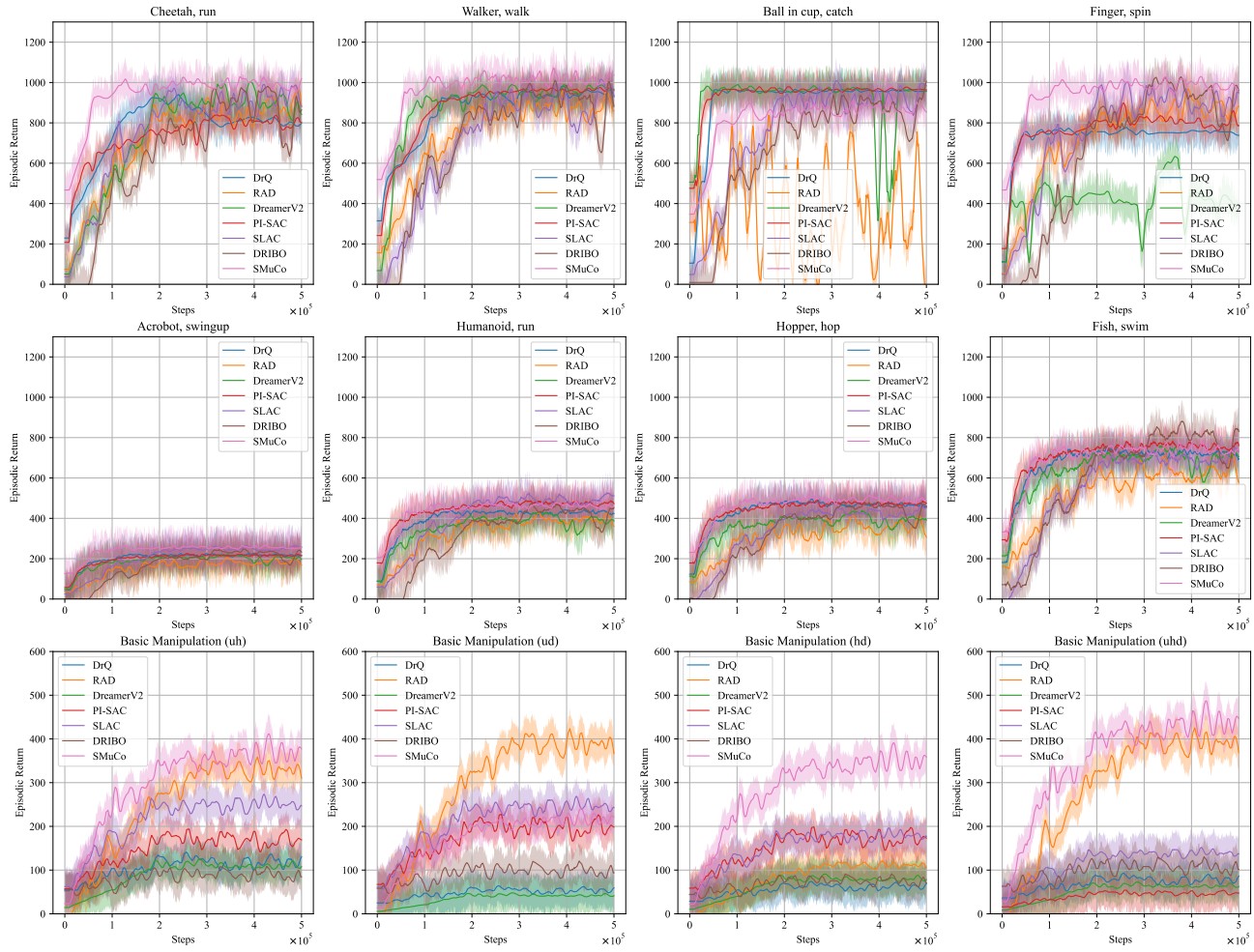

Figure 4: Evaluation on DMC tasks and basic manipulation task in Sapien. Row 1 shows results trained on DMC tasks: (Cheetah, run), (Walker, walk), (Ball in cup, catch), (Finger, spin), (Acrobat, swingup), (Humanoid, run), (Hopper, hop), (Fish, swim). Row 2 shows results trained on basic manipulation with different view settings: uh - upward and horizontal, ud - upward and diagonal, hd - horizontal and diagnoal, uhd - upward, horizontal and diagonal.

both DreamerV2 and PI-SAC achieve better performance and sample efficiency than SMuCo in the ball-in-cup catch task, even though SMuCo performs better than RAD, SLAC and DRIBO in convergence rate. In the finger spin task, SMuCo excels than other baselines in sample efficiency, except that SMuCo and DRIBO achieve almost the same score at the end of the evaluation. In the acrobat swingup task, SMuCo achieves almost equivalent performance as PI-SAC and SLAC. Although the final score of DRIBO is decent, its volatility makes it uncomparable with SMuCo, PI-SAC and SLAC. In the humanoid run task, SMuCo outperforms all the other baselines at the beginning of a relatively early time step. In the hopper hop task, only DRIBO has the almost comparable performance with SMuCo, with significantly larger variance than SMuCo. In the fish swim task, SMuCo's final score ranks third among other baselines. However, DRIBO with first rank has not reached a stable state and SLAC outperforms SMuCo only at the end of training stage,

indicating that DRIBO and SLAC have non-dominant advantage over SMuCo. In the basic manipulation task with both upward-horizontal views and horizontal-diagonal views settings, SMuCo outperforms other baselines. However, in the upward and diagonal views setting, RAD and SLAC have better final scores than SMuCo. This implies that the choice of different perspective could incur impact on the performance of training efficiency, which is partially consistent with conclusions in [Hsu et al., 2022].

In the basic manipulation task with upward-horizontal-diagonal views setting, SMuCo achieves best score over other baselines and itself across other two views settings.

The results listed in Figure 4 shows that our proposed method can have significant performance improvement in scenarios with the real multi-view data in application.

## 4.4 ABLATION STUDY

In the ablation study, we aim to investigate the setting of several components that can affect the overall performance of SMuCo, including (a) loss function form; (b) sequence length $T$; (c) number of views $V$; (d) data augmentation methods; (e) size of views. We conduct these experiments under the same visual control task: walker walk using algorithm SMuCo.

**Loss Function**. The different contributions of the three components of the objective function: reconstruction entropy term $\mathcal{L}_{\text{REC}}$, expected log likelihood term $\mathcal{L}_{\text{LL}}$ and temporal contrastive term $\mathcal{L}_{\text{TC}}$ are reported in this part. The evaluation results with different combination of the three terms are demonstrated in Figure 5(a).

We observe that (1) none of reconstruction entropy term $\mathcal{L}_{\text{REC}}$, expected log likelihood term $\mathcal{L}_{\text{LL}}$ and temporal contrastive term $\mathcal{L}_{\text{TC}}$ can achieve success in the task of walker walk. Intuitively, if the loss function only contains reconstruction entropy term $\mathcal{L}_{\text{REC}}$, relevant information for control task cannot be preserved in the representation $Z$ due to the embedding collapse phenomenon. If the loss function is in the form of either expected log likelihood term $\mathcal{L}_{\text{LL}}$ or temporal contrastive term $\mathcal{L}_{\text{TC}}$, the representation $Z$ would contain too much task-irrelevant information, which exacerbates the learning process of the control task. (2) However, when combining reconstruction entropy term $\mathcal{L}_{\text{REC}}$ and expected log likelihood term $\mathcal{L}_{\text{LL}}$, the final performance is better than that of each term, because reconstruction entropy term $\mathcal{L}_{\text{REC}}$ encourages the representation to be concise while expected log likelihood term $\mathcal{L}_{\text{LL}}$ guarantees the sufficiency of the representation. (3) In contrast, the other two combinations: reconstruction entropy term $\mathcal{L}_{\text{REC}}$ plus temporal contrastive term $\mathcal{L}_{\text{TC}}$ and expected log likelihood term $\mathcal{L}_{\text{LL}}$ plus temporal contrastive term $\mathcal{L}_{\text{TC}}$, do not have a noticeable improvement in final performance compared to single-term cases.

It suggests that temporal contrastive term $\mathcal{L}_{\text{TC}}$ can guarantee the robustness of the representation as the final performance of SMuCo is slightly better than the case of reconstruction entropy term $\mathcal{L}_{\text{REC}}$ plus expected log likelihood term $\mathcal{L}_{\text{LL}}$. In conclusion, reconstruction entropy term $\mathcal{L}_{\text{REC}}$ and expected log likelihood term $\mathcal{L}_{\text{LL}}$ are essential terms for obtaining performant representation while temporal contrastive term $\mathcal{L}_{\text{TC}}$ is a regularization term that can make the representation more robust.

**Sequence Length** $T$. To investigate the effect of different sequence lengths on the performance of the agent, we conduct experiments with different sequence lengths $T$, and the results are shown in Figure 5(b). We observe that as the sequence length $T$ increases, the performance of the agent also increases slightly. It means that SMuCo can learn better representations with the help of incorporating longer temporal dynamics. However, as $T$ increases from 15 to 20, the performance has not improved significantly compared to the improvement on smaller $T$ values, indicating that the marginal improvement brought about by adding more time steps to calculate total correlation will gradually disappear as $T$ increases. Intuitively, if we set $T$ to be a too large integer, observations from time steps with large time gap would be independent with each other, in which case the sequential total correlation can be separated into the sum of sequential total correlations on shorter sequences of observations, which makes it merely no benefit in learning temporal dynamics when adding extra time steps of observations in this case.

**Number of Views** $V$. To investigate whether providing more views $V$ can improve task performance, we conduct experiments with different number of views $V$ and depict the results in Figure 5(c). It is shown that as the number of views $V$ increases, the performance of agents increases slightly, weaker than the effect of sequence length $T$. We deduce that the SMuCo objective captures larger cross-view correlation during the optimization of the observation encoder, improving the robustness of the learned representation. However, too many views can only bring in too much redundant information and deteriorate the learning process. Therefore, it is beneficial to use multi-view data but not necessary to collect too many views in order to achieve a satisfactory performance in practice, considering the extra efforts demanded in data collection and computation for extra views.

**Data Augmentation Methods** We investigate the effects of using different data-augmentation methods to generate views. Figure 6(d) depicts the evaluation results under task: Walker, walk using different view-generation methods: Grayscale, Random Crop, Rotate and Color Jitter. The result suggests that Random Crop is more beneficial for improving performance of SMuCo, which is consistent with the conclusion from [Laskin et al., 2020a]. Intuitively, since SMuCo needs to optimize over discrepancy between different views, Gradyscale, Rotate and Color Jitter does not increase discrepancy among different views.

**Size of Views**. The size of view is correlated to the degree of partial information in each view. We conduct experiments using different size of views $[32, 64, 128, 256]$ as illustrated in Figure 6(e). The result implies that as the size of raw observation increases, the performance of SMuCo increases correspondingly with decreasing acceleration. This observation is coherent with the fact that the mutual information among different views is a submodular function [Krause et al., 2008, Nemhauser et al., 1978].

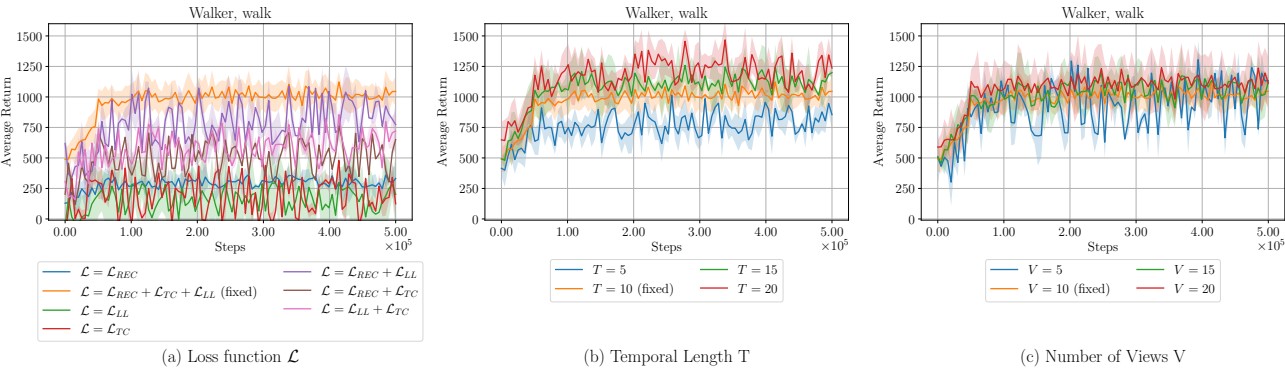

(a) Loss function $\mathcal{L}$      (b) Temporal Length T      (c) Number of Views V

Figure 5: Comparison among different settings.

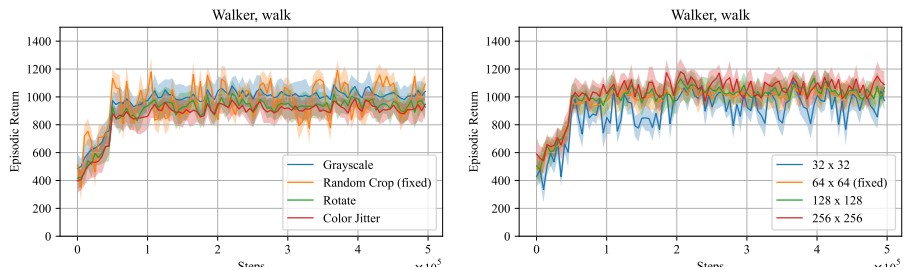

Figure 6: Evaluation results using different (left) data augmentation methods and (right) view sizes.

### 4.5 SPATIAL ATTENTION OF LEARNED REPRESENTATIONS

To show the representation learned by our proposed method, we build spatial attention maps of representations from observations [Zagoruyko and Komodakis, 2017], as illustrated in Figure 7. The SMuCo captures clear task-relevant information in the cheetah run and walker walk tasks (upper row in Figure 7) than the ball-in-cup catch and finger spin tasks (lower row in Figure 7). More specifically, the edges of the controlled agent can be clearly identified in the learned representation of cheetah run and walker walk, which also validates the good performance of SMuCo on these two tasks. Moreover, the learned representation in the finger spin task not only captures the current state of the agent, but also captures the possible future states as the movement would change the area around the agent in the view, which shows that SMuCo can capture the temporal dynamics in the learned representation.

## 5 CONCLUSION

In this work, we introduce a novel reinforcement learning algorithm named SMuCo specifically designed for visual control problems. SMuCo aims to learn a comprehensive and succinct representation of multi-view observations. By capturing shared information across views and exploiting temporal correlation, our approach maximizes sequential total

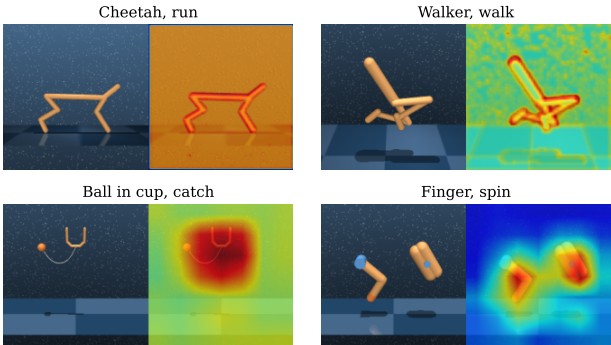

Figure 7: Spatial attention maps of representations from observations in four DMC tasks.

correlation between sequences of multi-view observations and their corresponding representations. To integrate temporal dynamics, we extend multi-view total correlation into sequential multi-view total correlation, conditioning on sequences of actions, and utilize it as the training objective for the encoder. In empirical evaluations, our proposed method demonstrates consistently superior performance compared to state-of-the-art baselines, including both model-free and model-based reinforcement learning methods.

**Limitations:** There are a few limitations for the SMuCo method. One crucial aspect revolves around the scalability concerns inherent in handling multiple views simultaneously. As outlined in [Hwang et al., 2021], the complexity of correlations across multiple views can significantly impact

the scalability of the method. This complexity not only poses challenges but can also exacerbate the difficulty of effectively learning and representing task-relevant information. These scalability limitations are particularly exacerbated when dealing with highly complex and scalable multi-view scenarios, such as those encountered in real-world applications. Consequently, while SMuCo may excel in certain contexts, its effectiveness and performance may be hindered in scenarios that demand handling many views and intricate correlations among them.

**Future Work:** As part of future work, SMuCo can be enhanced to handle challenges posed by unaligned multi-view observations and extend its capabilities to accommodate multi-modal observations, including not only image data but also text and audio data. This expansion will contribute to the algorithm's versatility across diverse input modalities in various applications. Furthermore, we could also try to address scenarios with limited multi-view data. One potential solution to this problem is to explore multi-view representation learning methodologies that can effectively handle such limitations. A promising avenue for this exploration is the CPM-Nets proposed by [Zhang et al., 2019]. CPM-Nets are designed to handle the absence or missingness of multi-view data, showcasing their effectiveness in dealing with data limitations. Therefore, our future work could focus on adapting and extending the principles of CPM-Nets to the domain of reinforcement learning, particularly in situations where there is a scarcity of multi-view information. By leveraging the benefits and methodologies of CPM-Nets, we could potentially develop novel approaches that are robust and efficient in learning policies despite limited multi-view data availability. This direction is promising for advancing reinforcement learning algorithms and addressing challenges posed by data constraints in multi-view environments.

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

# SMuCo: Reinforcement Learning for Visual Control via Sequential Multi-view Total Correlation
## (Supplementary Material)

## A  APPENDIX

### A.1  DETAILS ABOUT SMUCO SURROGATE OBJECTIVE

In this subsection, we elaborate on the details of derivation of the lower bound of sequential multi-view total correlation. Using this lower bound as a surrogate objective, we propose a reinforcement learning framework for visual control problems by learning a complete and concise representation from multi-view observations. We aim to maximize sequential multi-view total correlation between multi-view observation sequence and representation sequence under the condition of action sequence. First of all, we give a formal definition of sequential total multi-view correlation.

**Definition A.1.** *Given the sequence of multiview observations, representations, and actions $\vec{O}_{1:T}$, $Z_{1:T}$, $A_{1:T}$, define the sequential multiview total correlation as follows:*

$$SMTC(\vec{O}_{1:T}; Z_{1:T} \mid A_{1:T}) = \sum_{v=1}^{V} I(O_{1:T}^v; Z_{1:T} \mid A_{1:T}) - I(\vec{O}_{1:T}; Z_{1:T} \mid A_{1:T}) \tag{12}$$

*where $T$ is temporal length and $V$ is number of views.*

With the definition of sequential multi-view total correlation, we have the following lemmas and theorem to derive a tractable lower bound of mutual information between sequences of observation and representation on condition of action sequence.

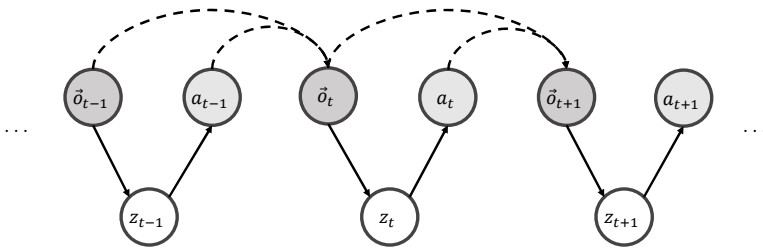

Figure 8: Graphical Model of Random Variables where $o$ denotes observation, $z$ denotes representation and $a$ denotes action.

**Lemma A.1.** *Let $O_{1:T}$ and $Z_{1:T}$ be random variables with joint distribution*

$$p(O_{1:T}, Z_{1:T}) = \prod_{t=1}^{T} p(O_t, Z_t \mid O_{t-1}, Z_{t-1}, A_{t-1})$$

*where $A_{1:T}$ be random variables of action sequence and $p(O_1, Z_1 \mid O_0, Z_0, A_0) = p(O_1, Z_1)$. Then it follows that*

$$I(O_{1:T}; Z_{1:T} \mid A_{1:T}) \geq \sum_{t=1}^{T} I(O_t; Z_t \mid Z_{t-1}, A_{t-1}) \tag{13}$$

*where $T$ is temporal length.*

*Proof.* According to information theory, we have

$$I(X;Y) = H(X) - H(X \mid Y),$$

$$H(X_1, X_2, \cdots, X_n) = \sum_{i=1}^{n} H(X_i \mid X_{i-1}, \cdots, X_1),$$

$$I(X;Y \mid Z) = \mathbb{E}_Z D_{KL}(P_{(X,Y)|Z} \parallel P_{X|Z} \otimes P_{Y|Z})$$

Let $\tau = (o_{1:t-1}, z_{1:t-1}, a_{1:T})$, according to the definition of conditional mutual information, we have

$$I(O_t; Z_t \mid O_{1:t-1}, Z_{1:t-1}, A_{1:T}) = \int_\tau \int_{o_t} \int_{z_t} p(o_t, z_t \mid \tau) \log \frac{p(o_t, z_t \mid \tau)}{p(o_t \mid \tau) \cdot p(z_t \mid \tau)} dz_t do_t d\tau \tag{14}$$

With the Markovian property, i.e. hidden state or representation $z_t$ at time step $t$ is determined only by previous hidden state or representation $z_{t-1}$ and action $a$ at time step $t-1$, it follows $p(o_t, z_t \mid \tau) = p(o_t, z_t \mid z_{t-1}, a_{t-1})$. Applying this result to Equation 14 we obtain the following result.

$$I(O_t; Z_t \mid O_{1:t-1}, Z_{1:t-1}, A_{1:T}) = \int_\tau \int_{o_t} \int_{z_t} p(o_t, z_t \mid z_{t-1}, a_{t-1}) \log \frac{p(o_t, z_t \mid z_{t-1}, a_{t-1})}{p(o_t \mid z_{t-1}, a_{t-1}) \cdot p(z_t \mid z_{t-1}, a_{t-1})}$$
$$dz_t do_t d\tau = I(O_t; Z_t \mid Z_{t-1}, A_{t-1}) \tag{15}$$

Finally, we derive a lower bound of mutual information between subsequences of observations and representations as

$$I(O_{1:T}; Z_{1:T} \mid A_{1:T}) = H(Z_{1:T} \mid A_{1:T}) - H(Z_{1:T} \mid O_{1:T}, A_{1:T}) \tag{16}$$

$$= \sum_{t=1}^{t} H(Z_t \mid Z_{1:t-1}, A_{1:T}) - H(Z_t \mid Z_{1:t-1}, O_{1:T}, A_{1:T}) \tag{17}$$

$$= \sum_{t=1}^{T} I(O_{1:T}; Z_t \mid Z_{1:t-1}, A_{1:T}) \tag{18}$$

$$= \sum_{t=1}^{T} H(O_{1:T} \mid Z_{1:t-1}, A_{1:T}) - H(O_{1:T} \mid Z_t, Z_{1:t-1}, A_{1:T}) \tag{19}$$

$$= \sum_{t=1}^{T} \sum_{s=1}^{T} H(O_s \mid O_{1:s-1}, Z_{1:t-1}, A_{1:T}) - H(O_s \mid O_{1:s-1}, Z_t, Z_{1:t-1}, A_{1:T}) \tag{20}$$

$$= \sum_{t=1}^{T} \sum_{s=1}^{T} I(O_s; Z_t \mid O_{1:s-1}, Z_{1:t-1}, A_{1:T}) \tag{21}$$

$$\geq \sum_{t=1}^{T} I(O_t; Z_t \mid O_{1:t-1}, Z_{1:t-1}, A_{1:T}) \tag{22}$$

$$= \sum_{t=1}^{T} I(O_t; Z_t \mid Z_{t-1}, A_{t-1}) \tag{23}$$

Hence, we have that

$$I(O_{1:T}; Z_{1:T} \mid A_{1:T}) \geq \sum_{t=1}^{T} I(O_t; Z_t \mid Z_{t-1}, A_{t-1}) \tag{24}$$

where $T$ is temporal length.

$\square$

**Lemma A.2.** *Let $\vec{O}_{1:T}$ be multi-view observation sequence with dim $\vec{O} = V$ and temporal length $T$, it follows that*

$$I(\vec{O}_{1:T}; Z_{1:T} \mid A_{1:T}) \leq \sum_{t=1}^{T} \sum_{s=1}^{T} \mathbb{E}_{p(\vec{o}_s)} \left[ D_{KL}(p(z_t \mid \iota) \parallel r_\phi(z_t \mid \iota)) \right] \tag{25}$$

*where $\iota = (\vec{o}_{1:s-1}, z_{1:t-1}, a_{1:T})$ and prior distribution $r_\phi(z_t) \approx p(z_t)$ is an approximate distribution with $\phi$.*

*Proof.* It is easy to verify that

$$I(\vec{O}; Z) \leq \mathbb{E}_{p(\vec{o})}[D_{KL}(p(z \mid \vec{o}) \parallel r_\phi(z))] \tag{26}$$

since

$$I(\vec{O}; Z) = \mathbb{E}_{p(\vec{o},z)}\left[\log \frac{p(z \mid \vec{o})}{p(z)} \cdot \frac{r_\phi(z)}{r_\phi(z)}\right] = \mathbb{E}_{p(\vec{o})}D_{KL}(p(z \mid \vec{o}) \parallel r_\phi(z)) - D_{KL}(p(z) \parallel r_\phi(z))$$
$$\leq \mathbb{E}_{p(\vec{o})}[D_{KL}(p(z \mid \vec{o}) \parallel r_\phi(z))] \tag{27}$$

Using this technique , we could derive a upper bound of $I(\vec{O}^{1:T}; Z^{1:T} \mid A^{1:T})$ as follows:

$$I(\vec{O}_{1:T}; Z_{1:T} \mid A_{1:T}) = \sum_{t=1}^{T}\sum_{s=1}^{T} I(\vec{O}_s; Z_t \mid \vec{O}_{1:s-1}, Z_{1:t-1}, A_{1:T}) \leq \sum_{t=1}^{T}\sum_{s=1}^{T} \mathbb{E}_{p(\vec{o}_s)}[D_{KL}(\tilde{q} \parallel \tilde{r})] \tag{28}$$

where

$$\tilde{q} = p(z_t \mid \vec{o}_{1:s}, z_{1:t-1}, a_{1:T}) \tag{29}$$
$$\tilde{r}_\phi = r_\phi(z_t \mid \vec{o}_{1:s-1}, z_{1:t-1}, a_{1:T}). \tag{30}$$

Hence, we derive an upper bound of mutual information between sequence of multi-view observations and representations on condition of action sequence.

$\square$

**Lemma A.3.** *For any view $v$ and time step $t$, we have*

$$H(O_t^v \mid Z_t, Z_{t-1}, A_{t-1}) \leq 0 \tag{31}$$

*where $O_t^v$ is observation of view $v$ at time step $t$, $Z_t$ and $Z_{t-1}$ are representation at time steps $t$ and $t-1$, $A_{t-1}$ is action at time step $t-1$.*

*Proof.* According to the definition of entropy, we have

$$H(O_t^v \mid Z_t, Z_{t-1}, A_{t-1}) = -\int_{o_t^v}\int_{z_t}\int_{z_{t-1}}\int_{a_{t-1}} f(o_t^v, z_t, z_{t-1}, a_{t-1}) \log g(o_t^v \mid z_t, z_{t-1}, a_{t-1}) do_t^v dz_t dz_{t-1} da_{t-1} \tag{32}$$

where $f$ is joint probability density function and $g$ is conditional probability density function.

If there does not exist independent noise in observation with respect to representation, according to the formulation of episode rollout of Markov Decision Process, we assert that at time step $t$ representation $Z$ and multi-view observation $O_t^v$ are mutually determined. Since observation $o_t^v$ is fully determined by representation $z_t$, it follows that $f(o_t^v, z_t, z_{t-1}, a_{t-1}) = f(z_t, z_{t-1}, a_{t-1})$. Using this result, we could reduce the integral into

$$H(O_t^v \mid Z_t, Z_{t-1}, A_{t-1}) = -\int_{z_t}\int_{z_{t-1}}\int_{a_{t-1}} f(z_t, z_{t-1}, a_{t-1}) \left[\int_{o_t^v} \log g(o_t^v \mid z_t, z_{t-1}, a_{t-1}) do_t^v\right]$$
$$dz_t dz_{t-1} da_{t-1} = 0 \tag{33}$$

where the second equality holds since there exists only one observation instance $o_t^v$ in observation space which corresponds to the representation condition $z_t$. Otherwise, the probability density function $g$ is zero.

If there exists independent noise in observation with respect to representation, it follows that

$$H(O_t^v \mid Z_t, Z_{t-1}, A_{t-1}) = -\int_{z_t}\int_{z_{t-1}}\int_{a_{t-1}} f(z_t, z_{t-1}, a_{t-1}) \left[\int_{o_t^v} \log g(o_t^v \mid z_t, z_{t-1}, a_{t-1}) do_t^v\right] dz_t dz_{t-1} da_{t-1}$$
$$= -C\int_{z_t}\int_{z_{t-1}}\int_{a_{t-1}} f^2(z_t, z_{t-1}, a_{t-1}) dz_t dz_{t-1} da_{t-1} \leq 0 \tag{34}$$

where $C$ is a positive constant.

Specifically, given some view index $v$ and time step $t$, it follows from Bayesian Theorem that

$$Pr(O_t^v \mid Z_t) = \frac{Pr(Z_t \mid O_t^v) \cdot Pr(O_t^v)}{\int_{O'} Pr(Z_t \mid O') dO'}. \tag{35}$$

According to the encode mapping, we know that $Z \sim \mathcal{N}(\mu, \boldsymbol{\Sigma})$ where $\mu$ is mean vector and $\boldsymbol{\Sigma}$ is covariance matrix. The mean and covariance matrix are determined by multi-view observations, i.e. $\mathcal{N}(\mu, \boldsymbol{\Sigma}) = h(\vec{O})$ for some encoding mapping $h$. Note that an encoding mapping is mapping from observational space into representation space. In multi-view scenario, the observational space consists of multi-view observations instead of single-view observation.

According to graphical model illustrated in Figure 8 and Equation 35, it holds that

$$Pr(O_t^v \mid Z_t, Z_{t-1}, A_{t-1}) = Pr(Z_t, Z_{t-1}, A_{t-1}) \cdot Pr(O_t^v \mid Z_t) \cdot Pr(O_t^v \mid Z_{t-1}, A_{t-1}) \tag{36}$$

$$= \frac{Pr(Z_t \mid O_t^v) \cdot Pr(O_t^v)}{\int_{O'} Pr(Z_t \mid O') dO'} \cdot Pr(O_t^v \mid Z_{t-1}, A_{t-1}) \cdot Pr(Z_t, Z_{t-1}, A_{t-1}) \tag{37}$$

$$= \frac{Pr(Z_t \mid O_t^v) \cdot Pr(O_t^v)}{Pr(Z_t)} \cdot Pr(O_t^v \mid Z_{t-1}, A_{t-1}) \cdot Pr(Z_t, Z_{t-1}, A_{t-1}). \tag{38}$$

With this result, we could derive

$$\int_{o_t^v} \log g(o_t^v \mid z_t, z_{t-1}, a_{t-1}) do_t^v = f(z_t, z_{t-1}, a_{t-1}) \cdot \int_{o_t^v} \log \left( \frac{p_1(z_t \mid o_t^v) \cdot p_2(o_t^v)}{p_3(z_t)} \cdot p_4(o_t^v \mid z_{t-1}, a_{t-1}) \right) do_t^v$$

$$= (T_1 + T_2 - T_3 + T_4) \cdot f(z_t, z_{t-1}, a_{t-1}) \tag{39}$$

where

$$T_1 := \int_{o_t^v} \log p_1(z_t \mid o_t^v) do_t^v, \tag{40}$$

$$T_2 := \int_{o_t^v} \log p_2(o_t^v) do_t^v, \tag{41}$$

$$T_3 := \int_{o_t^v} \log p_3(z_t) do_t^v = \log p_3(z_t), \tag{42}$$

$$T_4 := \int_{o_t^v} \log p_4(o_t^v \mid z_{t-1}, a_{t-1}) do_t^v. \tag{43}$$

According to the definition of encoder architecture which includes a IVW structure, $p_1$ is multi-dimensional Gaussian distribution. If we assume that observation distribution, representation distribution and transition probability distribution are all Gaussian distribution which is consistent with model-based methods for previous works Hafner et al. [2021], Zhang et al. [2021]. Since $T_1 + T_2 - T_3 = 0$ and $T_4 \geq 0$ due to the property of transition probability, it follows that the aforementioned positive value $C$ does exist.

$\square$

**Theorem A.1.** *The sequential multi-view total correlation between sequences of multi-view observation and representation on condition of action sequence has the following lower bound:*

$$SMTC(\vec{O}_{1:T}; Z_{1:T} \mid A_{1:T}) \geq \sum_{v=1}^{V} \sum_{t=1}^{T} \left[ H(O_t^v \mid Z_{t-1}, A_{t-1}) + \mathbb{E}_{p(z_t, o_t^v \mid z_{t-1}, a_{t-1})} \ln q_\psi^v(o_t^v \mid z_t, z_{t-1}, a_{t-1}) \right]$$

$$- \sum_{t=1}^{T} \sum_{s=1}^{T} \mathbb{E}_{p(\vec{o}_s)} \left[ D_{KL}(p(z_t \mid o_s, \iota) \parallel r_\phi(z_t \mid \iota)) \right] \tag{44}$$

*where posterior distribution $q_\psi(o_t^v \mid z_t, z_{t-1}, a_{t-1}) \approx p(o_t^v \mid z_t, z_{t-1}, a_{t-1})$ is an approximate distribution with $\psi$, $\iota$ and $r_\phi$ are defined in Lemma A.2.*

*Proof.* Applying Equation 13 in Lemma A.1, we could derive

$$SMTC(\vec{O}_{1:T}; Z_{1:T} \mid A_{1:T}) = \sum_{v=1}^{V} I(O_{1:T}^v; Z_{1:T} \mid A_{1:T}) - I(\vec{O}_{1:T}; Z_{1:T} \mid A_{1:T}) \tag{45}$$

$$\geq \sum_{v=1}^{V} \sum_{t=1}^{T} I(O_t^v; Z_t \mid Z_{t-1}, A_{t-1}) - I(\vec{O}_{1:T}; Z_{1:T} \mid A_{1:T}) \tag{46}$$

Then by applying inequality 25 in Lemma A.2, we have

$$-I(\vec{O}_{1:T}; Z_{1:T} \mid A_{1:T}) \geq -\sum_{t=1}^{T} \sum_{s=1}^{T} \mathbb{E}_{p(\vec{o}_s)}[D_{KL}(p(z_t \mid \iota) \| r_\phi(z_t \mid \iota))] \tag{47}$$

where $\iota = (\vec{o}_{1:s-1}, z_{1:t-1}, a_{1:T})$.

We split the first summand $I(O_t^v; Z_t \mid Z_{t-1}, A_{t-1})$ into subtraction of two entropy terms and derive a lower bound using Lemma A.3 as follows.

$$I(O_t^v; Z_t \mid Z_{t-1}, A_{t-1}) = H(O_t^v \mid Z_{t-1}, A_{t-1}) - H(O_t^v \mid Z_t, Z_{t-1}, A_{t-1}) \geq H(O_t^v \mid Z_{t-1}, A_{t-1}) \tag{48}$$

According to Appendix A.1 Equation (11) and (12) in Hwang et al. [2021], we could further lower the double sum of expected value of KL divergence between true prior distribution $p$ and approximate prior distribution $r_\phi$.

$$-\sum_{t=1}^{T} \sum_{s=1}^{T} \mathbb{E}_{p(\vec{o}_s)}[D_{KL}(p(z_t \mid \iota) \| r_\phi(z_t \mid \iota))] \geq \sum_{v=1}^{V} \sum_{t=1}^{T} \mathbb{E}_{p(z_t, o_t^v \mid z_{t-1}, a_{t-1})} \ln q_\psi^v(o_t^v \mid z_t, z_{t-1}, a_{t-1})]$$
$$-\sum_{t=1}^{T} \sum_{s=1}^{T} \mathbb{E}_{p(\vec{o}^s)}[D_{KL}(p(z_t \mid \iota) \| r_\phi(z_t \mid \iota))] \tag{49}$$

Summing up inequalities 48 and 49, we could have a tractable lower bound of sequential multi-view total correlation between sequences of multi-view observations and representations on condition of action sequence as follows.

$$SMTC(\vec{O}_{1:T}; Z_{1:T} \mid A_{1:T}) \geq \sum_{v=1}^{V} \sum_{t=1}^{T} \left[ H(O_t^v \mid Z_{t-1}, A_{t-1}) + \mathbb{E}_{p(z_t, o_t^v \mid z_{t-1}, a_{t-1})} \ln q_\psi^v(o_t^v \mid z_t, z_{t-1}, a_{t-1}) \right]$$
$$-\sum_{t=1}^{T} \sum_{s=1}^{T} \mathbb{E}_{p(\vec{o}_s)} \left[ D_{KL}(p(z_t \mid o_s, \iota) \| r_\phi(z_t \mid \iota)) \right] \tag{50}$$

where $T$ is temporal length and $V$ is number of views. $\qquad\square$

## A.2 NETWORK ARCHITECTURES AND HYPERPARAMETERS

The hyperparameters is set empirically based on previous works, some of which are listed in Table 1. In particular, it is sufficient to confine $V$ and $T$ as small integers. Since $V$ and $T$ are small integers, the lower bound derived above is computationally efficient to approximate the sequential multi-view total correlation even though the form has double sum operator. In the following, we elaborate the designs of network architecture for encoder, decoder, actor and critic.

**Encoder Networks** The encoder architecture consists of three convolutional layers with $3 \times 3$ kernels, 32 channels, stride 2 and padding 1, just like the auto-encoder architecture. ReLU is applied after each convolution layer as activation function. After flattening the output of the last convolutional layer, this output is fed into a fully-connected layer, generating a 2048-dimensional feature vector. This feature vector is passed to another two fully-connected layer, bringing about 64-dimensional mean and 64-dimensional variance separately.

| Parameter | Value |
|---|---|
| learning rate | 0.001 |
| optimizer | Adam |
| number of views $V$ | 10 |
| temporal length $T$ | 10 |
| batch size | 256 |
| representation dimension | 64 |
| discount factor $\gamma$ | 0.99 |
| number of random seeds | 5 |

Table 1: Hyperparameters for SMuCo framework

**Decoder Networks** The decoder architecture starts with a fully-connected layer, transforming 64-dimensional representation $Z$ into 2048-dimensional feature vector. Then it consists of three transposed convolutional layers with $3 \times 3$ kernels, 32 channels, stride 2 and padding 1.

**Actor and Critic Networks** We follow the common implementation of SAC Haarnoja et al. [2018]. The actor and critic networks are implemented by MLPs with 256-dimensional hidden layers. However, the actor network has two different output layers, including mean output layer and variance output layer while the critic network has only one output layer with 1-dimension.

## A.3 ADDITIONAL EXPERIMENTAL RESULTS

| Scores at 500k Steps | DrQ | RAD | DreamerV2 | PI-SAC | SLAC | DRIBO | SMuCo |
|---|---|---|---|---|---|---|---|
| Cheetah, run | $797 \pm 116$ | $880 \pm 104$ | $841 \pm 57$ | $802 \pm 119$ | $881 \pm 116$ | $864 \pm 52$ | $\mathbf{1019 \pm 107}$ |
| Walker, walk | $930 \pm 46$ | $858 \pm 82$ | $966 \pm 117$ | $959 \pm 103$ | $930 \pm 107$ | $881 \pm 90$ | $\mathbf{1036 \pm 30}$ |
| Ball in cup, catch | $958 \pm 102$ | $-9 \pm 80$ | $955 \pm 82$ | $963 \pm 75$ | $983 \pm 98$ | $\mathbf{1006 \pm 39}$ | $853 \pm 115$ |
| Finger, spin | $738 \pm 79$ | $880 \pm 32$ | $366 \pm 105$ | $787 \pm 112$ | $947 \pm 58$ | $960 \pm 53$ | $\mathbf{969 \pm 52}$ |
| Acrobot, swingup | $228 \pm 50$ | $163 \pm 34$ | $209 \pm 106$ | $246 \pm 85$ | $\mathbf{256 \pm 45}$ | $242 \pm 67$ | $247 \pm 51$ |
| Humanoid, run | $470 \pm 60$ | $375 \pm 117$ | $436 \pm 104$ | $482 \pm 37$ | $453 \pm 117$ | $497 \pm 40$ | $\mathbf{507 \pm 105}$ |
| Hopper, hop | $454 \pm 89$ | $357 \pm 44$ | $347 \pm 52$ | $431 \pm 67$ | $453 \pm 41$ | $\mathbf{488 \pm 38}$ | $484 \pm 33$ |
| Fish, swim | $729 \pm 90$ | $546 \pm 114$ | $697 \pm 109$ | $694 \pm 102$ | $750 \pm 101$ | $\mathbf{787 \pm 38}$ | $748 \pm 114$ |
| Basic Manipulation (uh) | $130 \pm 30$ | $310 \pm 34$ | $108 \pm 39$ | $168 \pm 45$ | $246 \pm 34$ | $84 \pm 42$ | $\mathbf{377 \pm 45}$ |
| Basic Manipulation (ud) | $59 \pm 34$ | $\mathbf{366 \pm 32}$ | $41 \pm 38$ | $198 \pm 38$ | $242 \pm 42$ | $93 \pm 43$ | $221 \pm 41$ |
| Basic Manipulation (hd) | $68 \pm 45$ | $105 \pm 31$ | $78 \pm 33$ | $173 \pm 32$ | $177 \pm 33$ | $70 \pm 48$ | $\mathbf{358 \pm 42}$ |
| Basic Manipulation (uhd) | $86 \pm 47$ | $368 \pm 36$ | $62 \pm 40$ | $47 \pm 33$ | $136 \pm 30$ | $101 \pm 38$ | $\mathbf{446 \pm 34}$ |
| Scores at 100k Steps | DrQ | RAD | DreamerV2 | PI-SAC | SLAC | DRIBO | SMuCo |
| Cheetah, run | $723 \pm 102$ | $577 \pm 109$ | $532 \pm 77$ | $683 \pm 100$ | $632 \pm 49$ | $496 \pm 119$ | $\mathbf{940 \pm 76}$ |
| Walker, walk | $725 \pm 83$ | $651 \pm 100$ | $923 \pm 98$ | $826 \pm 70$ | $452 \pm 97$ | $532 \pm 81$ | $\mathbf{976 \pm 109}$ |
| Ball in cup, catch | $959 \pm 65$ | $543 \pm 35$ | $\mathbf{988 \pm 48}$ | $963 \pm 113$ | $612 \pm 114$ | $547 \pm 77$ | $813 \pm 68$ |
| Finger, spin | $744 \pm 36$ | $679 \pm 87$ | $479 \pm 30$ | $761 \pm 49$ | $715 \pm 114$ | $262 \pm 34$ | $\mathbf{932 \pm 78}$ |
| Acrobot, swingup | $216 \pm 61$ | $139 \pm 53$ | $187 \pm 111$ | $227 \pm 78$ | $158 \pm 33$ | $112 \pm 52$ | $\mathbf{230 \pm 103}$ |
| Humanoid, run | $435 \pm 77$ | $318 \pm 114$ | $399 \pm 60$ | $435 \pm 109$ | $266 \pm 32$ | $248 \pm 93$ | $\mathbf{479 \pm 47}$ |
| Hopper, hop | $403 \pm 84$ | $327 \pm 101$ | $333 \pm 86$ | $394 \pm 75$ | $274 \pm 82$ | $223 \pm 61$ | $\mathbf{456 \pm 91}$ |
| Fish, swim | $671 \pm 58$ | $444 \pm 104$ | $634 \pm 98$ | $651 \pm 118$ | $470 \pm 108$ | $376 \pm 101$ | $\mathbf{708 \pm 33}$ |
| Basic Manipulation (uh) | $103 \pm 46$ | $147 \pm 38$ | $54 \pm 46$ | $117 \pm 40$ | $187 \pm 42$ | $56 \pm 32$ | $\mathbf{234 \pm 31}$ |
| Basic Manipulation (ud) | $47 \pm 47$ | $174 \pm 30$ | $20 \pm 41$ | $137 \pm 37$ | $\mathbf{184 \pm 48}$ | $62 \pm 30$ | $137 \pm 31$ |
| Basic Manipulation (hd) | $54 \pm 33$ | $50 \pm 41$ | $39 \pm 40$ | $119 \pm 32$ | $134 \pm 44$ | $47 \pm 35$ | $\mathbf{222 \pm 47}$ |
| Basic Manipulation (uhd) | $68 \pm 44$ | $175 \pm 32$ | $31 \pm 45$ | $32 \pm 37$ | $104 \pm 43$ | $67 \pm 37$ | $\mathbf{277 \pm 46}$ |

Table 2: Evaluation scores at 100k/500k steps.

**Evaluation Scores at different Stages.** We also summarize the evaluation scores of SMuCo and other baselines at 100k and 500k steps in Table 2. In scores at 500k steps, SMuCo achieves the best performance among baselines in 7 tasks over 12 tasks. In scores at 100k steps, SMuCo achieves the best performance among baselines in 10 tasks over 12 tasks. This implies that SMuCo has significant advantage in sample efficiency over other baselines, even though its non-dominant performance

in scores at 500k steps.

**Predictability**. We provide predictions of future observations using the deconvolutional decoder in Figure 9, showing that our method can really have good predictive ability for future states.

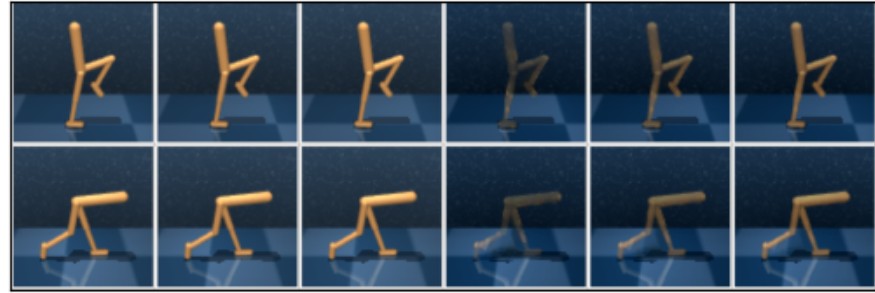

Figure 9: Prediction of future states (observations) from two representations of observations in task: Walker, walk.

## A.4    REASONS FOR POOR PERFORMANCE ON BALL CATCH

We posit that one of the possible reasons why SMuCo performs worse than other baselines in "Ball in cup, catch" is that too little difference exists among multiview observations under this scenario. Nevertheless, even though SMuCo may not be optimal for this single task, this does not impair the advantages over other baselines. Likewise, DRIBO Fan and Li [2022] does not outperform other baselines in finger spin. We would like to emphasize that SMuCo is not designed to outperform all existing model-free RL methods, but rather to introduce a novel approach to image representation learning and demonstrate its effectiveness in a set of benchmark tasks.

## A.5    SUMMARY OF VISUAL CONTROL METHODS

| Method | Model-free | Representation | Information theoretic |
|--------|:----------:|:--------------:|:---------------------:|
| SMuCo | ✓ | ✓ | ✓ |
| DreamerV2 Hafner et al. [2021] | ✗ | ✗ | ✗ |
| RAD Laskin et al. [2020a] | ✓ | ✗ | ✗ |
| DrQ Yarats et al. [2021] | ✓ | ✗ | ✗ |
| PI-SAC Lee et al. [2020b] | ✓ | ✓ | ✓ |
| SLAC Lee et al. [2020a] | ✓ | ✓ | ✗ |
| DRIBO Fan and Li [2022] | ✓ | ✓ | ✓ |

Table 3: Comparison among SMuCo and other baselines.