# OpenReview forum: "SMuCo: Reinforcement Learning for Visual Control via Sequential Multi-view Total Correlation"
_auai.org/UAI/2024/Conference — UAI 2024 poster_

### Official Review · Reviewer_Kqpq · 2024-03-20

**Q2-1 Originality-Novelty:** 2
**Q2-2 Correctness-Technical Quality:** 3
**Q2-5 Clarity Of Writing:** 3

**Q1 Summary And Contributions:**

The paper proposes a novel method to exploit multiple views for reinforcement learning. In particular, they introduce a novel observation encoder objective based on the sequential total correlation between sequences of multi-view observations and representations conditioned on action sequences.

**Q2-3 Extent To Which Claims Are Supported By Evidence:**

2: Fair: the main claims are somewhat supported by evidence (but the experimental evaluation may be weak, or does not match entirely with the claims, important baselines may be missing, proofs contain important ideas but lack rigor, algorithmic details are only discussed superficially, references are imprecise, assumptions are not sufficiently motivated or explicated, etc.).

**Q2-4 Reproducibility:**

2: Fair: key resources (e.g. proofs, code, data) are unavailable but key details (e.g. proof sketches, experimental setup) are sufficiently well-described for an expert to confidently reproduce the main results.

**Q3 Main Strengths:**

The paper deals with an interesting and challenging scenario in reinforcement learning. The experimental setup is greatly detailed. The evaluation compares SMuCo to SOTA baselines on challenging tasks, including a task where SMuCo fails. The comprehensive ablation study provides intriguing insights.

**Q4 Main Weakness:**

The introduction discusses the “robot arm catching problem”, where multiple views offer valuable insights to solve the task. However, for the experimental section, the views were generated through data augmentation such as gray scaling or random crop. I feel that this is a missed opportunity to showcase the advantages of having multiple views.

**Q5 Detailed Comments To The Authors:**

As described above, my main concern about the paper is the nature of views used for the experiments. I do not see how augmentations such as grey scaling provide more useful information than the original, unaugmented observation. This makes me wonder whether the impressive experimental results arise simply from using a larger quantity of data points, rather than more informative ones.

I believe there should be more explanation on how one arrives at Equation 6 from Equations 2 and 5. Is there a formal derivation for how this combination arises, or does Equation 5 serve only as inspiration and SMTC is derived from intuition?

I am confused by how the reward $r_t$ and the next multi-view observation $o_t$ is generated at lines 7 and 8 in Algorithm 1. I would assume that these variables come from the environment $E$ through $r_t = R(o_t, a_t)$ and $o_{t+1} \sim \mathcal P(\cdot | o_t, a_t)$. However, Algorithm 1 suggests that these variables depend on the representations $z_t$ provided by the learned encoder instead of $o_t$. Does this mean that the $R$ and $\mathcal P$ are world models? If $z_t$ represents the true state of the environment, then why is it sampled from the (not yet optimal) encoder?

I suspect line 8 in Algorithm 1 should obtain $o_{t+1}$ (used immediately in the next line) instead of $o_{t}$.

I am also confused by the update at line 12 in Algorithm 1. According to Equations 7,9,10 and 11, this update requires the multi-view observation $o_t$. However, it is not clear to me where this observation comes from at line 12. Is it the observation obtained at line 8, used to update the encoder several times, or does it come from the replay buffer $\mathcal D$? In that case, shouldn’t Algorithm 1 also put $o_t$ into $\mathcal D$ at line 10?

If space requirements allow, Figures 4 and especially 5 could be greatly improved with a larger font size, better matching the font size of the text body. For Figure 4, consider using a single shared legend.

I believe there is a typo on page 4: “remove task-**ir**relevant information from learned representations”

**Q9 Complying With Reviewing Instructions:**

Yes

---

> ### Author Rebuttal · Authors · 2024-04-06
>
> While we acknowledge that data augmentation methods like random crop may not completely replicate the advantages of genuine multiple views, there are valid reasons why we chose to utilize them as substitutes in our research. Firstly, during the conduct of this study, the DeepMind Control Suite did not offer built-in methods to generate multi-view observations. Despite this limitation, the Control Suite remains a robust testbed for comparing our methods with baselines, as many baseline algorithms have been evaluated using this task set. Even if SMuCo doesn't achieve optimal performance, it's essential to demonstrate its relative advantages over baselines, ensuring fairness in experimental design and facilitating meaningful comparisons. Secondly, even if we could access ideal multi-view observations within the DeepMind Control Suite, it would be unfair to exclusively feed these multi-view pixels into SMuCo. Doing so might provide SMuCo with a larger amount of information compared to other baselines, potentially skewing the experimental results and introducing bias. Thirdly, the multiple views generated by these data augmentation methods do not violate the assumption that task-relevant information is preserved in all views. Therefore, by employing data augmentation methods, we ensure a level playing field where each algorithm receives a similar amount of information, promoting fair comparisons and unbiased evaluation of performance across different methods.
>
> Equations 2 and 5 serve only as inspiration for constructing SMTC. Indeed, one may come up with other similar metrics. But it may be hard to derive a tractable surrogate objective to optimize.
>
> We greatly thank the reviewer to point out the following typos.
>
> **Potential Typos to be fixed:**
> 1. We are sorry for the confusion at line 7 and 8. The reward and next multi-view observation are dependent with previous multi-view observation, not representations. Hence, $R$ and $\mathcal{P}$ are not world models.
> 2. In addition, the subscript of next observation at line 8 should be $t+1$, not $t$.
> 3. For line 12, it indeed use the same sampling technique from replay buffer to compute the surrogate objective and its gradient through backward propogation. Hence, tuple in line 10 should also include mutli-view observation $\vec{o}_t$.
> 4. In page 4, the objective is to remove task-irrelevant, not relevant, information from learned representations.
>
> For the resolution of experimental figure, we will try our best to increase the readability of results.

---

### Official Review · Reviewer_ekAy · 2024-03-23

**Q2-1 Originality-Novelty:** 3
**Q2-2 Correctness-Technical Quality:** 3
**Q2-5 Clarity Of Writing:** 3

**Q1 Summary And Contributions:**

This paper introduces SMuCo, an innovative algorithm for visual control tasks within reinforcement learning (RL) systems. SMuCo utilizes inputs from multiple viewpoints to capture identical physical states and constructs robust latent representations by optimizing the sequential total correlation across multiple views, addressing the challenge of integrating multi-view data into representation learning. Extensive experiments in two public environments demonstrate SMuCo's superiority over model-free and model-based baseline RL algorithms.
This paper proposes a novel reinforcement learning framework that leverages multi-view total correlation to advance representation learning in visual control tasks. Additionally, This paper develops the SMuCo objective. This objective characterizes the multi-view total correlation between observation sequences and their representations, efficiently capturing crucial temporal dynamics relevant to the task, and discarding out non-essential details.

**Q2-3 Extent To Which Claims Are Supported By Evidence:**

2: Fair: the main claims are somewhat supported by evidence (but the experimental evaluation may be weak, or does not match entirely with the claims, important baselines may be missing, proofs contain important ideas but lack rigor, algorithmic details are only discussed superficially, references are imprecise, assumptions are not sufficiently motivated or explicated, etc.).

**Q2-4 Reproducibility:**

3: Good: key resources (e.g. proofs, code, data) are available and key details (e.g. proofs, experimental setup) are sufficiently well-described for competent researchers to confidently reproduce the main results.

**Q3 Main Strengths:**

1. SMuCo's approach to handling visual control tasks with sequential multi-view is innovative. Most existing work focuses on single-view sequences, but a single view cannot fully reflect the information related to the task, thus it fails to fully capture the potential state representations in the images.
2. SMuCo excels in building robust latent representations through the optimization of multi-view sequential total correlation. It effectively gathers essential task-related information and temporal dynamics, simultaneously disregarding irrelevant data. This capability allows SMuCo to support an extensive range of views and showcases its versatility and adaptability in various visual control tasks involving multi-view data.
3. The experiments are extensive and sufficient hyper-parameters are provided for reproduction.

**Q4 Main Weakness:**

1. While leveraging multi-view data is an advantage of SMuCo, it also becomes a weakness in scenarios where such data is not easily obtainable or is costly to acquire. In situations where it is challenging to obtain multi-view data that provides complementary information, or the quality of multi-view data is low, the effectiveness of the method in learning robust representations may be compromised.
2. The method's detailed focus on capturing task-relevant information and temporal dynamics while filtering out irrelevant data could lead to overfitting, especially in cases where the diversity of the training data is insufficient to represent the complexity of real-world environments. This potential risk may limit the model's generalizability and effectiveness in unseen scenarios.
3. The method's reliance on optimizing multi-view sequential total correlation increases the complexity of the model. This complexity may hinder the method's application in large datasets or real-world application scenarios that require substantial computational resources.

**Q5 Detailed Comments To The Authors:**

This article presents SMuCo, an innovative algorithm in Reinforcement Learning (RL) systems for visual control tasks. The article has areas for improvement:
1. SMuCo focuses intensely on capturing relevant information and temporal dynamics for tasks, but filtering out irrelevant data might cause overfitting.
2. The model's complexity might restrict its use in practical scenarios with large datasets.
3. The authors should explore how well the model performs and its suitability in cases with limited multi-view data.

**Q9 Complying With Reviewing Instructions:**

Yes

---

> ### Author Rebuttal · Authors · 2024-04-06
>
> **Responses for Q4**
> 1. We agree with points 1 and 3 in the Q4 Main Weakness section. Firstly, if the quality of multi-view data is relatively low, the performance of SMuCo is not guaranteed. For example, in our empirical evaluation, SMuCo failed to outperform baselines due to the minimal gap between the multi-view data generated in the “Ball in cup, catch” control task. Secondly, the complexity of SMuCo increases linearly with the number of views and quadratically with the length of the episode. In this case, SMuCo is not suitable for application scenarios with a high number of views and extensive temporal structure concurrently. These two points represent the limitation of our method, and we will summarize and include them in the conclusion section.
> 2. Overfitting is a common issue in machine learning where a model learns the training data too well, to the extent that it negatively impacts its ability to generalize to new, unseen data. In control tasks used in our experiments, background pixels are not relevant to the optimization objective of Markov Decision Process. If simulation environment could give parameters of current physical state, it would be more efficient to learn a good policy using this kind of concise state. However, in visual control, the primary obstacle is caused by redundant information in comparison with the former scenario. Hence, optimizing the removal of redundant task-irrelevant information does not lead to overfitting. Furthermore, the current target of visual control problems is to learn a performant agent with some algorithm given a specification of problem since visual pixels is hard to learn using traditional methods like Naive Policy Gradient or Q Learning. The generalization of our policy is not our prioritized goal in the design of SMuCo. Nevertheless, it is meaningful to extend SMuCo to achieve better generalization power in the future work, where we need to split environment set into training and testing instance sets.
> 3. See point 1.
>
> **Responses for Q5**
> 1. See point 2 in Q4.
> 2. See point 1 in Q4.
> 3. We thank the reviewer for this great suggestion, and we will consider this direction as future work. One potential solution to address scenarios with limited multi-view data is to explore multi-view representation learning methodologies that can effectively handle such limitations. A promising avenue for this exploration is the CPM-Nets proposed by *Changqing Zhang et al. (2019) CPM-Nets: Cross Partial Multi-View Networks*. CPM-Nets are designed to handle the absence or missingness of multi-view data, showcasing their effectiveness in dealing with data limitations. Therefore, our future work could focus on adapting and extending the principles of CPM-Nets to the domain of reinforcement learning, particularly in situations where there is a scarcity of multi-view information. By leveraging the benefits and methodologies of CPM-Nets, we could potentially develop novel approaches that are robust and efficient in learning policies despite limited multi-view data availability. This direction is promising for advancing reinforcement learning algorithms and addressing challenges posed by data constraints in multi-view environments.

---

### Official Review · Reviewer_ZjU2 · 2024-03-23

**Q2-1 Originality-Novelty:** 3
**Q2-2 Correctness-Technical Quality:** 3
**Q2-5 Clarity Of Writing:** 3

**Q1 Summary And Contributions:**

This paper aims to address the challenge of learning effective representations in visual RL tasks by integrating temporal dynamics into multi-view total correlation. Key contributions include extending multi-view total correlation into sequential multi-view total correlation, and proposing a novel representation learning method for visual reinforcement learning based on this.

**Q2-3 Extent To Which Claims Are Supported By Evidence:**

2: Fair: the main claims are somewhat supported by evidence (but the experimental evaluation may be weak, or does not match entirely with the claims, important baselines may be missing, proofs contain important ideas but lack rigor, algorithmic details are only discussed superficially, references are imprecise, assumptions are not sufficiently motivated or explicated, etc.).

**Q2-4 Reproducibility:**

3: Good: key resources (e.g. proofs, code, data) are available and key details (e.g. proofs, experimental setup) are sufficiently well-described for competent researchers to confidently reproduce the main results.

**Q3 Main Strengths:**

The paper introduces a novel method that integrates temporal dynamics into multi-view total correlation to learn effective representations in visual reinforcement learning (RL), supported by both theoretical and empirical evidence. Additionally, it is clearly written and seems to be easy to reproduce.

**Q4 Main Weakness:**

The experimental evaluation may not match entirely with the claims. The authors emphasized their proposed method can capture task-relevant information while filtering out irrelevant data. However, they do not perform experiments on tasks with obvious task-irrelevant distractions.

**Q5 Detailed Comments To The Authors:**

The paper is well-written and the authors have not only performed ablation study on the proposed method but also validated the proposal on several benchmark tasks. However, there are some concerns to be resolved at this stage based on the following comments.
(1) The authors emphasized their proposed method can capture task-relevant information while filtering out irrelevant data, but they do not perform experiments on tasks with obvious task-irrelevant distractions. The authors can consider to further verify their method’s ability to learn task-relevant information on such tasks, e.g., Distracting DeepMind Control Suite.
(2) The author can improve the resolution of the images to increase clarity.
(3) The limitations of the new proposed method should be mentioned in the conclusion section.

**Q9 Complying With Reviewing Instructions:**

Yes

---

> ### Author Rebuttal · Authors · 2024-04-06
>
> (1) We greatly appreciate the valuable suggestion. The attention map depicted in Figure 7 indirectly clarifies that our focus is solely on the task-relevant parts. Therefore, the presence of a cluttered background may not significantly impact the performance of our proposed method, as our attention is directed towards the relevant areas and not the background.
>
> (2) We will increase the resolution of the images as soon as possible.
>
> (3) We will append discussion of limitations in conclusions. **Limitations**: The limitations of the SMuCo method are multifaceted, particularly when considering the challenges posed by the multi-view sequential total correlation approach. One crucial aspect revolves around the scalability concerns inherent in handling multiple views simultaneously. As outlined in *Hwang et al. (2021) Multi-View Representation Learning via Total Correlation Objective*, the complexity of correlations across multiple views can significantly impact the scalability of the method. This complexity not only poses challenges but can also exacerbate the difficulty of effectively learning and representing task-relevant information. These scalability limitations are particularly exacerbated when dealing with highly complex and scalable multi-view scenarios, such as those encountered in real-world applications. Consequently, while SMuCo may excel in certain contexts, its effectiveness and performance may be hindered in scenarios that demand handling many views and intricate correlations among them.

---

### Official Review · Reviewer_2phs · 2024-03-24

**Q2-1 Originality-Novelty:** 2
**Q2-2 Correctness-Technical Quality:** 3
**Q2-5 Clarity Of Writing:** 3

**Q1 Summary And Contributions:**

The paper introduces a reinforcement learning algorithm called SMuCo, specifically designed for visual control problems. This algorithm learns a latent representation of multi-view observations by maximizing the overall correlation of sequential multi-views. SMuCo supports an unlimited number of views and demonstrates superior performance over leading model-free and model-based RL algorithms, which were validated by empirical experimental results.

**Q2-3 Extent To Which Claims Are Supported By Evidence:**

2: Fair: the main claims are somewhat supported by evidence (but the experimental evaluation may be weak, or does not match entirely with the claims, important baselines may be missing, proofs contain important ideas but lack rigor, algorithmic details are only discussed superficially, references are imprecise, assumptions are not sufficiently motivated or explicated, etc.).

**Q2-4 Reproducibility:**

3: Good: key resources (e.g. proofs, code, data) are available and key details (e.g. proofs, experimental setup) are sufficiently well-described for competent researchers to confidently reproduce the main results.

**Q3 Main Strengths:**

1. The paper addresses a practical and challenging problem of effectively utilizing multi-view data in reinforcement learning. Such a problem is critical.

2. The paper provides enough experimental results to support the effectiveness of the proposed method. In multiple standard DMC environments, the SMuCo algorithm demonstrates superior performance compared to the current state-of-the-art methods.

**Q4 Main Weakness:**

1. Despite SMUCO’s impressive performance across multiple tasks, its performance on certain specific tasks is not as strong as other baseline methods. The authors should give some possible reasons for this.

2. The paper lacks a discussion on algorithm complexity and computational efficiency. These factors are equally important for practical applications.

3. The idea of constructing robust latent representations from multiple-view data is common and trivial. The authors might be better to emphasize the differences between other baselines and explain why using sequential total correlation could obtain higher performances.

2. Why not consider Fuse2Control (F2C) [Hwang et al., 2023] as a baseline method?

**Q5 Detailed Comments To The Authors:**

Please see Q4.

**Q9 Complying With Reviewing Instructions:**

Yes

---

> ### Author Rebuttal · Authors · 2024-04-06
>
> 1. In Appendix Sec. A.4, we have provided a potential explanation for the bad performance observed in the “Ball in cup, catch” task. The visual pixels comprising this task include a curved cup, a thin string, and a small ball. With the current methods used for view generation, there is minimal differentiation among the multi-view data, thereby undermining the potential advantages offered by multi-view total correlation methods compared to other baselines. However, we are confident that if visual pixels from diverse angles, creating sufficient disparities among the multi-view data, can be accessed, the benefits of SMuCo could potentially become evident.
> 2. We will append discussion of limitations in conclusions. **Limitations**: The limitations of the SMuCo method are multifaceted, particularly when considering the challenges posed by the multi-view sequential total correlation approach. One crucial aspect revolves around the scalability concerns inherent in handling multiple views simultaneously. As outlined in *Hwang et al. (2021) Multi-View Representation Learning via Total Correlation Objective*, the complexity of correlations across multiple views can significantly impact the scalability of the method. This complexity not only poses challenges but can also exacerbate the difficulty of effectively learning and representing task-relevant information. These scalability limitations are particularly exacerbated when dealing with highly complex and scalable multi-view scenarios, such as those encountered in real-world applications. Consequently, while SMuCo may excel in certain contexts, its effectiveness and performance may be hindered in scenarios that demand handling many views and intricate correlations among them.
> 3. We have elaborated the comparison between baselines and our methods in Sec. 4.2. DreamerV2 sets itself apart by integrating a world model to understand agent behaviors, explicitly preserving latent dynamics. In contrast, other techniques like RAD and DrQ do not explicitly model dynamics. Both RAD and DrQ input transformed observations, raw pixels from interactions with the environment, into downstream reinforcement learning (RL) tasks. On the contrary, SMuCo considers the joint representation from encoding multi-view observations as the state, emphasizing task-relevant information for downstream RL tasks. In the case of RAD and DrQ, a notable distinction lies in observational transformation. RAD employs data augmentation techniques such as color jittering and random cropping, while DrQ samples transformation operators from an invariant state transformation set, referring to this approach as a data regularized method. Furthermore, PI-SAC achieves representation learning by maximizing a Conditional Entropy Bottleneck (CEB)-related surrogate as an auxiliary task for training the encoder. In contrast, DRIBO aims to maximize the mutual information between two marginal representations and the divergence of likelihood probability. These differences underscore the varied approaches and methodologies each method employs in the field of representation learning for RL. One of the possible reasons why SMuCo could outperform other baselines lies in the sufficiency and conciseness of representation obtained by sequential total correlation surrogate.
> 4. Considering Fuse2Control (*Hwang et al. (2023) Information-Theoretic State Space Model for Multi-View Reinforcement Learning*) as a baseline method was not possible during the development of our work because F2C had not been published at that time. Our research was conducted independently and predates the availability of F2C as a reference. Additionally, even if F2C had been available earlier, there are fundamental differences in the modeling assumptions between F2C and our method. F2C assumes a Partially Observable Markov Decision Process (POMDP) model and incorporates State Space Models (SSM), while our method does not rely on these assumptions. This difference in modeling approaches makes direct comparison challenging, as our method aims to provide a more flexible and generalizable solution without being constrained by specific modeling assumptions. Therefore, while F2C is acknowledged as a related work, our decision not to consider it as a baseline method is based on both timing considerations and fundamental differences in modeling strategies.

---

### Official Review · Reviewer_smLE · 2024-03-28

**Q2-1 Originality-Novelty:** 3
**Q2-2 Correctness-Technical Quality:** 3
**Q2-5 Clarity Of Writing:** 4

**Q1 Summary And Contributions:**

The paper introduces a new algorithm SMuCo, an innovative multi-view reinforcement learning algorithm that constructs robust latent representations by optimizing multiview sequential total correlation. This technique effectively captures task-relevant information and
temporal dynamics while filtering out irrelevant data.

**Q2-3 Extent To Which Claims Are Supported By Evidence:**

3: Good: the main claims are supported by convincing evidence (in the form of adequate experimental evaluation, proofs, (pseudo-)code, references, assumptions).

**Q2-4 Reproducibility:**

2: Fair: key resources (e.g. proofs, code, data) are unavailable but key details (e.g. proof sketches, experimental setup) are sufficiently well-described for an expert to confidently reproduce the main results.

**Q3 Main Strengths:**

A well written and explained papers. Figures are visualising interesting and well thought out.

**Q4 Main Weakness:**

Provide a link to the code used to generate the paper for reproducibility.

(see below).

**Q5 Detailed Comments To The Authors:**

Please explain in the introduction / literature by explaining how you came to your methodology from your research questions, this will further strengthern the paper.

Also, how did you gather this methodology could be suitable given the data you had seen?
Sec 3.1, Explain how you modified the existing technique and the theoretical guarantees of the multi-view total correlation

Explain more in th conclusions the limitation of this view for the temporal setup, note this quote: "it can make the problem itself harder depending on the complexity of correlations across all views and the scalability of handling a number of views." From this paper in your references: HyeongJoo Hwang, Geon-Hyeong Kim, Seunghoon Hong, and Kee-Eung Kim. Multi-view representation learning
via total correlation objective; especially in line with this quote from your appendix: "We would like to emphasize that SMuCo is not designed to outperform all existing model-free RL methods".

Describe more how the The Total Correlation method has been used in AI even outside representation learning in the literature review.

**Q9 Complying With Reviewing Instructions:**

Yes

---

> ### Author Rebuttal · Authors · 2024-04-06
>
> **Methodology**: The development of our methodology is guided by three important research questions and insights gained from prior studies. Firstly, another crucial research question is about the challenges involved in effectively extracting task-relevant information from images while filtering out irrelevant details during the representation learning process. This challenge is identified through references like *Amy Zhang et al. (2021) Learning Invariant Representations for Reinforcement Learning without Reconstruction* and *Jiameng Fan et al. (2022) DRIBO: Robust Deep Reinforcement Learning via Multi-View Information Bottleneck*, which highlight the challenges of this task. Secondly, the introduction also emphasizes the increasing availability and usefulness of multi-view data in various application scenarios. This leads to the exploration of how additional perspectives from multi-view data can contribute to distinguishing task-relevant information from irrelevant details, independent of specific actions. Thirdly, the introduction also outlines the desired attributes of the learned representation, such as its ability to predict future states based on potential actions while discarding task-irrelevant visual details. This likely guides the formulation of the SMuCo objective, which focuses on preserving task-relevant temporal dynamics while eliminating task-irrelevant information. By addressing these research questions, SMuCo is formulated to tackle three problems, leverage multi-view data, and enhance performance in visual control tasks through skilled representation learning.
>
> **Modification and Derivation**: Previous work *Hwang et al. (2021) Multi-View Representation Learning via Total Correlation Objective* has proposed a representation learning method using the multi-view total correlation technique. However, this method may not be suitable for reinforcement learning in visual control, as the reinforcement learning problem involves sequential decision-making and temporal structures significantly influence the performance of the policy in reinforcement learning algorithms. To leverage the advantages of multi-view learning effectively, appropriate modifications need to be made to the multi-view total correlation method. These modifications should enable it to achieve optimality across different views and utilize the temporal structure inherent in Markov decision processes. While there are several possible ways to incorporate temporal structure into multi-view total correlation, the conditional entropy bottleneck technique stands out due to its promising theoretical guarantee as a surrogate loss for optimization.
>
> **Limitations**: The limitations of the SMuCo method are multifaceted, particularly when considering the challenges posed by the multi-view sequential total correlation approach. One crucial aspect revolves around the scalability concerns inherent in handling multiple views simultaneously. As outlined in *Hwang et al. (2021) Multi-View Representation Learning via Total Correlation Objective*, the complexity of correlations across multiple views can significantly impact the scalability of the method. This complexity not only poses challenges but can also exacerbate the difficulty of effectively learning and representing task-relevant information. These scalability limitations are particularly exacerbated when dealing with highly complex and scalable multi-view scenarios, such as those encountered in real-world applications. Consequently, while SMuCo may excel in certain contexts, its effectiveness and performance may be hindered in scenarios that demand handling many views and intricate correlations among them.
>
> **Extension for literature review of total correlation methods**: Total correlation, a fundamental measure derived from information theory, plays a pivotal role beyond representation learning *Chen et al. (2018), Locatello et al. (2019), Kim and Mnih (2018)* within the realm of AI, finding application across a diverse spectrum of tasks. In the context of independent component analysis (ICA), *Cardoso (2003)* introduces a comprehensive framework that establishes connections between mutual information, entropy, and non-Gaussianity, all without relying on decorrelation constraints. This framework contributes significantly to understanding the underlying structures within complex datasets by leveraging the inherent dependencies among variables. Moving to the domain of structure discovery, *Ver Steeg and Galstyan (2014)* proposes a novel methodology centered around learning a hierarchical structure of progressively abstract representations of intricate data sets. This approach is underpinned by optimizing an information-theoretic objective, ensuring that the learned representations capture meaningful and salient features of the data while facilitating interpretability and scalability.

---

### Meta-Review · Area_Chair_pcQw · 2024-04-20

The paper describes a new technique for multi-view RL.  This represents an important advance for visual control.  The reviewers unanimously recommend acceptance.  The authors are encouraged to improve the clarity of the derivations based on the reviewers' comments and to include additional experiments where the multiple views arise from truly different viewpoints as opposed to augmentations that do not include more information.